# The cytochrome *b* carboxyl terminal region is necessary for mitochondrial complex III assembly

Daniel Flores-Mireles[1], Yolanda Camacho-Villasana[1] , Madhurya Lutikurti[2], Aldo E García-Guerrero[3] , Guadalupe Lozano-Rosas[4], Victoria Chagoya[4], Emma Berta Gutiérrez-Cirlos[5] , Ulrich Brandt[2], Alfredo Cabrera-Orefice[2] , Xochitl Pérez-Martínez[1] 

Mitochondrial *bc₁* complex from yeast has 10 subunits, but only cytochrome *b* (Cyt*b*) subunit is encoded in the mitochondrial genome. Cyt*b* has eight transmembrane helices containing two hemes *b* for electron transfer. Cbp3 and Cbp6 assist Cyt*b* synthesis, and together with Cbp4 induce Cyt*b* hemylation. Subunits Qcr7/Qcr8 participate in the first steps of assembly, and lack of Qcr7 reduces Cyt*b* synthesis through an assembly-feedback mechanism involving Cbp3/Cbp6. Because Qcr7 resides near the Cyt*b* carboxyl region, we wondered whether this region is important for Cyt*b* synthesis/assembly. Although deletion of the Cyt*b* C-region did not abrogate Cyt*b* synthesis, the assembly-feedback regulation was lost, so Cyt*b* synthesis was normal even if Qcr7 was missing. Mutants lacking the Cyt*b* C-terminus were non-respiratory because of the absence of fully assembled *bc₁* complex. By performing complexome profiling, we showed the existence of aberrant early-stage subassemblies in the mutant. In this work, we demonstrate that the C-terminal region of Cyt*b* is critical for regulation of Cyt*b* synthesis and *bc₁* complex assembly.

## Introduction

Mitochondrial complex III or *bc₁* complex is part of the respiratory chain. It oxidizes ubiquinol through the "Q cycle" and reduces cytochrome *c*. Coupled to this redox process, *bc₁* complex contributes to the electrochemical gradient of protons across the inner membrane, which is used by the $F_1F_O$–ATP synthase (complex V) to produce ATP. In yeast and mammals, the *bc₁* complex has 10 and 11 subunits, respectively, of which cytochrome *b* (Cyt*b*) is the only subunit encoded in the mitochondrial genome. Cyt*b*, together with subunits cytochrome *c₁* (Cyt*c₁*), and the Rieske iron–sulfur protein

(Rip1) contain the redox centers, whereas the rest of subunits have structural roles (Schägger et al, 1995). Mutations that affect function, stability or assembly of this enzyme are causes of severe mitochondrial pathologies in humans (Ghezzi et al, 2011; Meunier et al, 2013; Wanschers et al, 2014).

Assembly of the *bc₁* complex is a modular process. In yeast, Cyt*b* forms an early assembly intermediate complex together with subunits Qcr7 and Qcr8. Subunits Cor1, Cor2, and Cyt*c₁* form another assembly intermediate. These intermediates further associate, and together with subunit Qcr6, they form a 500 kD intermediate that is already dimerized (Zara et al, 2007, 2009b; Conte et al, 2015; Stephan & Ott, 2020). Finally, subunits Qcr9, Qcr10, and Rip1 are added to form the functional enzyme (Zara et al, 2009a; Ndi et al, 2018). In addition, the *bc₁* complex associates with cytochrome *c* oxidase (complex IV) to form respiratory supercomplexes (Cruciat et al, 2000; Schägger & Pfeiffer, 2000; Hartley et al, 2019).

Cyt*b* is a hydrophobic subunit of the *bc₁* complex, with eight transmembrane helices and two hemes *b* of low ($b_L$) and high ($b_H$) redox potential that are directly involved in electron transfer. This subunit is encoded by in the mitochondrial genome, synthesized in the mitochondrial matrix, and inserted into the inner membrane. Current knowledge indicates that in yeast, translational activation of the *COB* mRNA depends on Cbp1, Cbs1, and Cbs2 (Dieckmann et al, 1984; Rödel, 1986; Rödel & Fox, 1987; Islas-Osuna et al, 2002). These proteins act on the *COB* 5′-UTR mRNA and interact with the mitoribosome to allow translation initiation, probably tethering translation to the mitochondrial inner membrane (Gruschke et al, 2012; Ott et al, 2016). Cbs1 might act as a translational repressor maintaining the *COB* mRNA close to the ribosomal exit tunnel, and only after translation activation, Cbs1 is released from the mitoribosome (Salvatori et al, 2020). Cbp3 and Cbp6 are chaperones that interact with the mitoribosome tunnel exit and with the newly synthesized Cyt*b* (Gruschke et al, 2011). In some yeast strains, Cbp3/Cbp6 are necessary for *COB* mRNA-efficient translation (Dieckmann & Tzagoloff, 1985; Wu & Tzagoloff, 1989; Gruschke et al, 2011, 2012;

[1]Departamento de Genética Molecular, Instituto de Fisiología Celular, Universidad Nacional Autónoma de México, México City, México [2]Radboud Institute for Molecular Life Sciences, Radboud University Medical Center, Nijmegen, The Netherlands [3]Department of Medicine and Biochemistry, University of Utah School of Medicine, Salt Lake City, UT, USA [4]Departamento de Biología Celular y Desarrollo, Instituto de Fisiología Celular, Universidad Nacional Autónoma de México, México City, México [5]Laboratorio de Bioquímica y Bioenergética, UBIMED, FES Iztacala, UNAM, México City, México

Correspondence: xperez@ifc.unam.mx; alfredbiomed@gmail.com

García-Guerrero et al, 2018), although the main role of these proteins is to trigger heme $b_L$ addition to Cyt$b$ through a mechanism that is not completely understood (Hildenbeutel et al, 2014; García-Guerrero et al, 2018). The current model states that interaction of Cyt$b$, Cbp3 and Cbp6 form intermediate 0 (Hildenbeutel et al, 2014). Upon addition of the heme $b_L$ site, the chaperone Cbp4 (Crivellone, 1994) is recruited to stabilize the hemylated Cyt$b$ forming intermediate I, comprising Cyt$b$, Cbp3, Cbp6, and Cbp4 (Hildenbeutel et al, 2014). After addition of heme $b_H$ by an unknown mechanism, Cbp3 and Cbp6 are released, whereas subunits Qcr7 and Qcr8 are then added to form intermediate II. This intermediate is ready to associate with the Cor1/Cor2/Cyt$c_1$ subcomplex to continue the assembly pathway (Zara et al, 2007, 2009b; Stephan & Ott, 2020). The current model proposes that in mutants where complex III assembly is blocked, Cbp3 and Cbp6 are sequestered in intermediates containing Cyt$b$ making them unavailable for more rounds of *COB* mRNA translation. This assembly-feedback regulation of translation is clearly observed in mutants lacking subunit Qcr7 or deficient in Cyt$b$ hemylation (Gruschke et al, 2012; Hildenbeutel et al, 2014).

High resolution structures of complex III from different organisms show that the carboxyl terminal region of Cyt$b$ is facing the mitochondrial matrix and in close proximity to the Qcr7 amino terminal region (Iwata et al, 1998; Lange & Hunte, 2002; Sousa et al, 2016; Wu et al, 2016; Guo et al, 2017; Rathore et al, 2019; Berndtsson et al, 2020; Hartley et al, 2020). This region could have an important mechanistic role for Cbp3/Cbp6 function during Cyt$b$ synthesis and assembly, because the release of Cbp3/Cbp6 from assembly intermediates has only been described before the association of Qcr7 with Cyt$b$ to form intermediate II. Cbp3/Cbp6 and Qcr7 could thus interact with the same region of the Cyt$b$ C-terminus. To investigate the possible role of the Cyt$b$ C-terminal region in complex III biogenesis, we deleted its last 8 or 13 residues. We demonstrate that this critical region of Cyt$b$ is essential for synthesis regulation by assembly-feedback, and for correct progression of $bc_1$ complex assembly. We also detected the presence of aberrant assembly subcomplexes that accumulate in the absence of the Cyt$b$ C-terminal region.

## Results

### The Cyt$b$ carboxyl terminal region is necessary for correct regulation of Cyt$b$ synthesis and respiratory growth

According to the available high resolution structures of the mitochondrial $bc_1$ complex from different species, the Cyt$b$ C-terminus is facing the matrix side and interacts with subunit Qcr7 (Iwata et al, 1998; Lange & Hunte, 2002; Sousa et al, 2016; Wu et al, 2016; Guo et al, 2017; Rathore et al, 2019; Berndtsson et al, 2020; Hartley et al, 2020). In fact, Cyt$b$, Qcr7, and Qcr8 are the first subunits to interact during the $bc_1$ complex assembly (Zara et al, 2009b; Gruschke et al, 2012). Compared with other complex III subunits, absence of Qcr7 has the greatest impact on decreasing Cyt$b$ synthesis (Gruschke et al, 2012; Garcia-Guerrero et al, 2018). For this reason, we hypothesized that the Cyt$b$ C-terminal region could play a central role in regulation of Cyt$b$ synthesis. Furthermore, subunit Qcr7 was proposed to properly

bind to Cyt$b$ only after Cbp3/Cbp6 are released (Zara et al, 2009b; Gruschke et al, 2012), thus the Cyt$b$ C-terminal region might be a major contact site shared by those proteins. To investigate whether the Cyt$b$ C-terminal region is critical for Cyt$b$ synthesis, we truncated it by deleting several residues. By microprojectile bombardment, we inserted two different mutant versions of *COB* in the mitochondrial genome. In the Cyt$b$ΔC13 mutant, we deleted the sequence IENVLFYIGRVNK comprising the last 13 amino acids of the protein. This region is exposed to the mitochondrial matrix and contains the sequence IEN that is highly conserved among fungi and mammals (Fig S1A). We also created the Cyt$b$ΔC8 mutant, where the last 8 amino acids (FYIGRVNK) were eliminated. In the fully assembled complex III, the somewhat less conserved FYI sequence of the Cyt$b$ C-terminal region resides closest to the N-terminal region of Qcr7.

In both strains, truncation of Cyt$b$ induced a complete lack of respiratory growth when grown in the presence of non-fermentable carbon sources like ethanol/glycerol and lactate (Fig 1A), similar to a control strain where the *COB* mRNA translational activator Cbs1 was deleted (Rödel, 1986; Rödel & Fox, 1987). To test whether truncation of the Cyt$b$ C-terminus abolished complex III assembly by inhibiting Cyt$b$ synthesis, we performed in vivo mitochondrial translation assays in the presence of ($^{35}$S)-methionine and cycloheximide (to inhibit cytosolic translation). After 15 min of labeling, mitochondria were extracted and newly made polypeptides were separated by SDS–PAGE, transferred to a PVDF membrane, and analyzed by autoradiography. Synthesis of mitochondrial products, including Cyt$b$ synthesis, was unchanged in Cyt$b$ΔC13 and Cyt$b$ΔC8 mutants as compared with WT (Fig 1B). In contrast, steady state levels of Cyt$b$ΔC13 and Cyt$b$ΔC8 were drastically reduced when analyzed by Western blot (Fig 1C), indicating that even though newly synthesized Cyt$b$ΔC13 and Cyt$b$ΔC8 were produced at WT levels, the truncated proteins exhibited markedly decreased stability.

In some WT yeast strains (e.g., BY4742 and W303), efficient translation of the *COB* mRNA depends on the chaperones Cbp3 and Cbp6, and on the presence of subunit Qcr7 (Gruschke et al, 2012; García-Guerrero et al, 2018). We asked whether these factors still regulate Cyt$b$ΔC13 synthesis, because the C-terminal region could be an important site of interaction for Cyt$b$ synthesis regulators. We generated deletion strains lacking Cbp3, Cbp6 or Qcr7 in cells carrying either WT Cyt$b$ or Cyt$b$ΔC13. As expected, in vivo mitochondrial translation assays revealed that WT Cyt$b$ synthesis was dramatically reduced in *Δcbp3*, *Δcbp6*, and *Δqcr7* mutants (Fig 1D). Synthesis of Cyt$b$ΔC13 was similarly reduced in the absence of Cbp3 and Cbp6. In contrast, Cyt$b$ΔC13 synthesis was no longer dependent on the presence of Qcr7, as the *Δqcr7* mutant showed normal levels of Cyt$b$ΔC13 ($^{35}$S)-methionine labeling. Similar results were obtained for the mutant Cyt$b$ΔC8 (Fig S1B). Cbp3 and Cbp6 are associated with the mitoribosome (Gruschke et al, 2011), although the exact role of this Cbp3/Cbp6 population on translation is not well understood. We tested this interaction by centrifugation of mitochondrial extracts on a sucrose cushion and analyzed the presence of Cbp3 in the ribosomal fraction. The Cbp3–mitoribosome interaction was unchanged by the presence of the *Cytb$\Delta$C13* mutation (Fig 1E). These results indicate that even though translation of the *COBΔC13* or *COBΔC8* mRNAs still depended on Cbp3 and Cbp6, it was

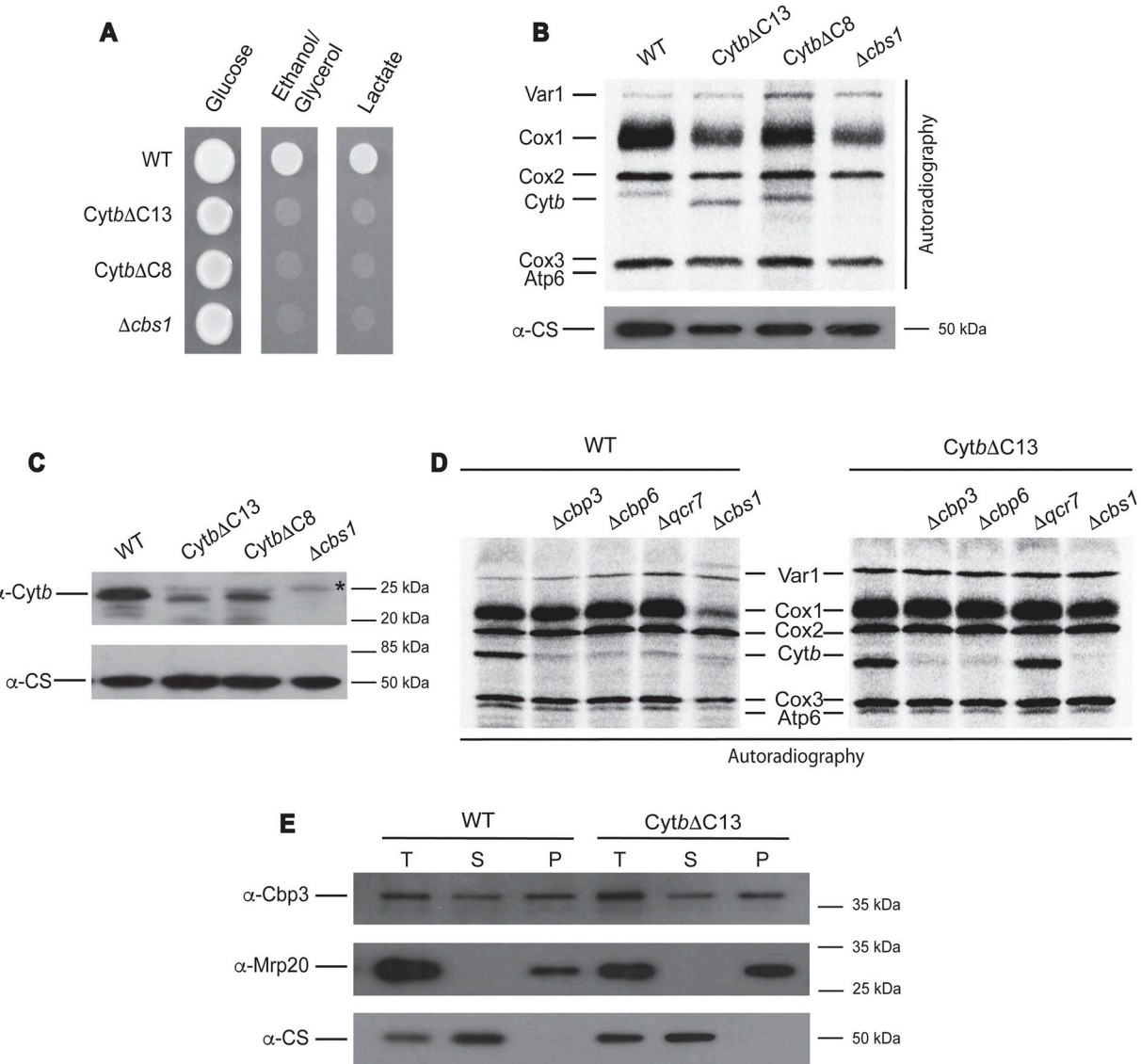

**Figure 1. The Cyt*b* C-terminal region is necessary for respiratory growth and Cyt*b* stability.**
**(A)** Cells carrying the indicated mutations were grown on complete media plates with glucose or ethanol/glycerol as carbon source for 2 d at 30°C. **(B)** Cells carrying the indicated mutations were labeled with (³⁵S)-methionine in the presence of cycloheximide and analyzed by SDS–PAGE, Western blot, and autoradiography. Mitochondrial products were Var1 (small ribosomal subunit protein Var1); Cox1, Cox2, and Cox3, subunits 1, 2, and 3, respectively, from cytochrome *c* oxidase; Cyt*b*, subunit from *bc₁* complex; Atp6, subunit 6 from ATP synthase. The *Δcbs1* mutant was used as a negative control for the absence of Cyt*b* synthesis (Rödel & Fox, 1987). Citrate synthase (CS) was detected by Western blot and autoradiography and was used as loading control. **(C)** A sample of 20 µg of mitochondrial protein from the indicated strains was loaded on SDS–PAGE and analyzed by Western blot. Citrate synthase (CS) antibody was used as loading control. This is a representative Western blot experiment from three independent repeats. The asterisk (*) indicates an unspecific signal associated with the polyclonal antibody against Cyt*b*. **(D)** Cells carrying WT Cyt*b* or Cyt*b*ΔC13 were labelled with (³⁵S)-methionine in the presence of cycloheximide as in Fig 1B. **(C, E)** Mitochondria (300 µg) from WT and the Cyt*b*ΔC13 mutant were lysed with 1% digitonin. 50% of the sample was analyzed as total fraction (T) and the rest was ultracentrifuged on a 1.2 M sucrose cushion. Supernatant (S) and pellet-ribosomal (P) fractions were TCA precipitated and analyzed by Western blot with antibodies against Cbp3, citrate synthase CS (a soluble protein), and Mrp20 (component of the mitoribosomal large subunit).

no longer regulated by Qcr7 binding and consequently by the assembly-feedback mechanism.

According to the current model (Ott et al, 2016), Cbp3/Cbp6 are translational activators of the *COB* mRNA. If complex III assembly is blocked at early stages (e.g., by the absence of Qcr7), then Cbp3/Cbp6 remain associated with Cyt*b*, so these chaperones are not available for *COB* mRNA translational activation. This model explains why *Δqcr7* mutants show reduced Cyt*b* synthesis (Gruschke et al, 2012; García-Guerrero et al, 2018). Therefore, we next investigated whether Cyt*b*ΔC13 and Cyt*b*ΔC8 still interact with Cbp3 by using a strain expressing Cbp3 with a hemagglutinin epitope fused to its C-terminal region (Cbp3-HA). The presence of this tag does not affect the respiratory capacity of the cells (García-Guerrero et al, 2018). Mitochondria were isolated and subjected to

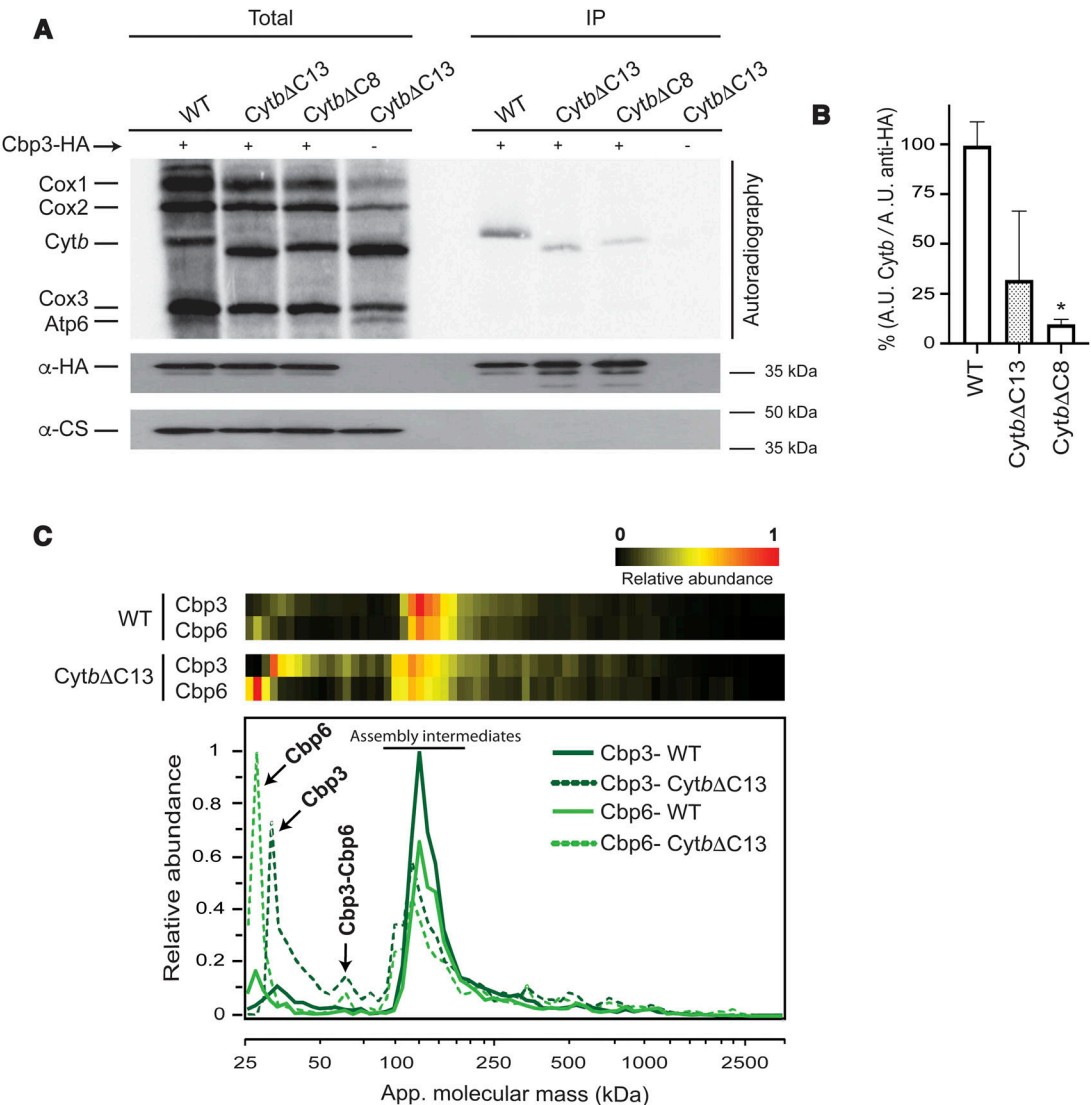

**Figure 2. The Cyt*b* C-terminal region is necessary for assembly-feedback regulation of Cyt*b* synthesis.**
**(A)** Purified mitochondria (500 µg) from the indicated strains and carrying a hemagglutinin-tagged Cbp3 (Cbp3-HA) were labeled with ($^{35}$S)-methionine. 10% of the reaction was used for the total fraction. Mitochondria were solubilized with 1% digitonin, and extracts were immunoprecipitated with a HA antibody. Samples were separated by SDS–PAGE and transferred to a PVDF membrane. This membrane was analyzed by autoradiography and incubated with the indicated antibodies. Citrate synthase (CS) antibody was used as loading control and as negative control for immunoprecipitation of Cbp3-HA. **(B)** Averaged densitometries of immunoprecipitated Cyt*b* were normalized to the signal from the immunoprecipitated Cbp3-HA and expressed as mean ± SD of three different repetitions (n = 3). The resulting averaged quotient from the WT strain was set as 100% and the averaged values from the mutant strains were adjusted accordingly. The densitometries were obtained using ImageJ. Statistical analysis performed by one-way ANOVA followed by Bonferroni correction to compare the mean of each group to the control (WT), *$P < 0.05$ versus WT. **(C)** Mitochondria (100 µg) were solubilized with digitonin (3 mg/mg protein) and separated by BN-PAGE. One lane from each sample was cut into 60 slices and analyzed by complexome profiling. The heatmaps (top) and abundance plots (bottom) show the relative abundance of Cbp3 and Cbp6 (data normalized to maximum iBAQ values across the two profiles). Averaged data from three biological replicates are shown.

($^{35}$S)-methionine labeling before solubilization with digitonin. Extracts were immunoprecipitated with an anti-HA commercial antibody. Samples were separated by SDS–PAGE and analyzed by autoradiography and Western blotting. Newly synthesized Cyt*b*ΔC13 and Cyt*b*ΔC8 were able to physically interact with Cbp3-HA, although with less efficiency, as was consistently observed in all our repeats. However, only the reduction in the Cyt*b*ΔC8 mutant showed statistical significance (Fig 2A and B). Because strains carrying Cyt*b*ΔC13 and Cyt*b*ΔC8 had presented highly similar results, we

decided to continue our experiments only with the Cyt*b*ΔC13 mutant.

Considering that Cbp3 seemed to have a weaker interaction with Cyt*b*ΔC13 than with full-length Cyt*b*, and Cyt*b*ΔC13 synthesis was no longer feedback regulated, we asked whether the assembly intermediates containing Cbp3/Cbp6 were modified in our mutant. mitochondria were solubilized with digitonin and separated, first by blue native PAGE (BN-PAGE), and then by 2D SDS–PAGE. As expected, WT Cyt*b* mitochondria showed the presence of Cbp3 at ~150–440 kD

corresponding to early complex III assembly intermediates (Gruschke et al, 2012) (Fig S2). A small fraction of Cyt$b$ was also detected in these subcomplexes, but the main population was observed at fully assembled complex III dimers and super-complexes. In contrast, Cyt$b\Delta$C13 no longer comigrated with supercomplexes or complex III$_2$. Instead, it was only detected in subcomplexes of an apparent mass of ~130–440 kD. A population of Cbp3 comigrated with these subcomplexes. In addition, Cbp3 was enriched in a fraction migrating at ~60 kD, which corresponds to the Cbp3–Cbp6 heterodimer. The ~30 kD fraction corresponding to free Cbp3 was hardly detectable (Gruschke et al, 2012). In the absence of Cyt$b$ ($\Delta cbs1$ mutant), Cbp3 was observed only as Cbp3/Cbp6 heterodimer and monomer in similar intensities, with no other complex detected. These results suggest that the absence of the Cyt$b$ C-terminal region induced accumulation of the Cbp3–Cbp6 dimer, formation of additional Cbp3-containing complexes of higher molecular mass, and a severe reduction of steady state levels of Cyt$b\Delta$C13.

To better define the populations of Cbp3- and Cbp6-containing complexes that are present in the Cyt$b\Delta$C13 mutant, we carried out a complexome profiling analysis. For this, purified mitochondria were solubilized with digitonin, and extracts were separated by BN-PAGE. Gel lanes from the control and mutant were cut into 60 slices, and each fraction was trypsin-digested and analyzed by LC–MS/MS. Cbp3 and Cbp6 had similar migration patterns, with a predominant population around 100–200 kD (Fig 2C). Indeed, one main peak with a clear shoulder at ~140 kD was observed in the WT Cyt$b$ strain from both proteins. These fractions correspond to the previously observed early-assembly intermediates (Gruschke et al, 2012). In the Cyt$b\Delta$C13 mutant, these peaks were decreased by ~50%, with the appearance of an additional shoulder at lower mass (~100 kD). The previously described Cbp3/Cbp6 complex of 66 kD (Gruschke et al, 2012) was hardly detectable in WT and enriched in Cyt$b\Delta$C13 mitochondria. The apparent mass of ~60 kD for the heterodimeric complex fitted well with its expected molecular mass of 53 kD. Monomeric Cbp3 and Cbp6 were found to accumulate in the mutant mitochondria as well.

Taken together, these results indicate that the C-terminal region of Cyt$b$ is critical for the assembly-feedback mechanism of Cyt$b$ synthesis. The increased free fractions and heterodimers of Cbp3/Cbp6 probably accumulated because of their lower affinity to Cyt$b\Delta$C13 and thus were released from assembly intermediates, kept Cyt$b$ synthesis going, but without proper regulation. This lack of regulation might be because of the presence of increased concentrations of Cbp3/Cbp6 heterodimers or free Cbp3 and Cbp6 in the mutant. Moreover, Cyt$b\Delta$C13 led to the formation of additional subcomplexes of smaller size containing Cbp3 and Cbp6.

### The Cyt$b$ carboxyl terminal region is indispensable for complex III assembly

Although we found that translation of the *COB$\Delta$C13* and *COB$\Delta$C8* mRNAs was as efficient as that of WT *COB* mRNA, Cyt$b$ steady state levels decreased in the mutants, respiratory capacity was lost completely, indicating severe complex III deficiency (Fig 1). Moreover, we observed that the formation of Complex III dimers and supercomplexes was abolished in the Cyt$b\Delta$C13 mutant (Fig S2). To

further investigate this, we first analyzed differential redox spectra of Cyt$b\Delta$C13 mitochondria. Although signals of heme $c$ and heme $a$ were unaffected, the heme $b$ peak at ~560 nm was absent in the Cyt$b\Delta$C13 mutant, similar to what was observed for a mutant lacking Cyt$b$ ($\Delta cbs1$) (Fig S3A). Besides, no complex III activity was detected in this mutant, which further confirms the loss of a functional assembled enzyme (Fig S3B). No cytochrome $c$ reduction catalyzed by individual complex III (using decylubiquinol as electron donor) or by coupling CII+CIII (using succinate to generate endogenous ubiquinol via CII to promote CIII activity) was detected in the mutant further confirming the loss of a functional assembled enzyme (Fig S3B). Individual activities of complexes II and IV were not significantly affected in this strain; hence, indicating a specific impairment at the complex III assembly level.

Next, we analyzed the behavior of the $bc_1$ complex subunits in more detail using the complexome profiling data. Whereas in WT mitochondria, all complex III subunits were primarily comigrating at ~500–1,000 kD (corresponding to supercomplexes [III$_2$/IV and III$_2$/IV$_2$] and complex III dimer), in Cyt$b\Delta$C13 mitochondria, no subunits were detectable in this mass range (Fig 3A). In addition, small amounts of both, Cyt$b\Delta$C13 and WT Cyt$b$, were detected in early-assembly intermediates at ~100–200 kD (see also Fig 4). In both WT and Cyt$b\Delta$C13 strains, a slight accumulation of a soluble sub-complex containing Cor1 and Cor2 was observed at ~300 kD (soluble mass scale), which fits well with a previously reported hetero-tetrametric arrangement, that is, (Cor1/2)$_4$ (Stephan & Ott, 2020). As expected, the formation of supercomplexes with complex IV (CIV) was dramatically reduced in Cyt$b\Delta$C13 mitochondria (Fig 3B). However, the expression of mitochondrially encoded COX subunits and enzyme activity of CIV were not affected in this mutant (Figs 1B and S3B). The relative content of the fully assembled complex was also unchanged, as evaluated by in gel CIV activity staining (Fig S3C).

### Lack of the Cyt$b$ C-terminal region leads to accumulation of aberrant early-stage subassemblies of $bc_1$ complex

So far, we have presented evidence that the Cyt$b$ C-terminal region is not essential for translation of the *COB* mRNA but is required for correct assembly of the $bc_1$ complex. The notion that this is because of its involvement in the assembly feedback regulation of Cyt$b$ synthesis was corroborated by our observation that Qcr7, a component of the first assembly intermediates, was not required for the stability of the truncated Cyt$b$. Cbp3 and Cbp6 have an essential role in the first steps of $bc_1$ complex assembly, which together with chaperone Cbp4 and subunit Qcr8, orchestrate the formation of the early intermediates coined 0 (Hildenbeutel et al, 2014), I, and II (Gruschke et al, 2011, 2012). To further corroborate our hypothesis of a feedback function of the Cyt$b$ C-terminal region, we asked how these early-stage intermediates were affected by its truncation. To address this, we analyzed subassemblies found at 50–200 kD mass range that contain Cyt$b$, Cbp3, Cbp6, Cbp4, Qcr7, and Qcr8.

Three different subassemblies have been described previously in this mass range: intermediate 0, containing Cyt$b$, Cbp3, and Cbp6; intermediate I, containing Cyt$b$, Cbp3/6, and Cbp4; and intermediate II, containing Cyt$b$, Cbp4, Qcr7, and Qcr8 (Gruschke et al, 2012; Hildenbeutel et al, 2014). Although we could confirm the formation of such intermediates as discussed below, our analysis provided

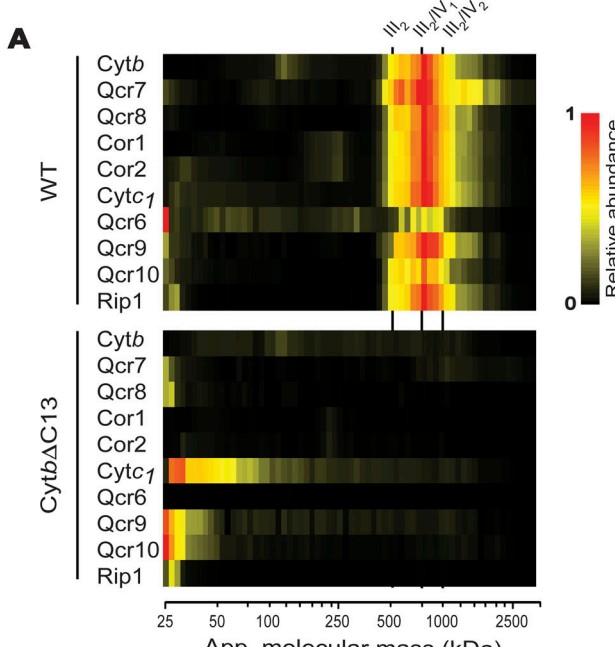

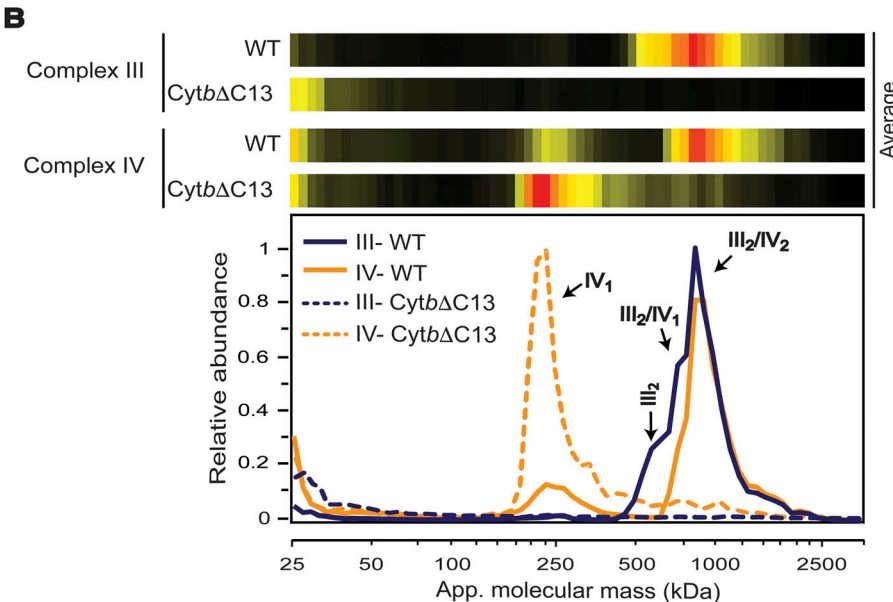

**Figure 3. The Cyt*b* carboxyl terminal region is necessary for assembly of the *bc₁* dimer and supercomplexes.**

**(A)** Heat-maps of migration profiles for all *bc₁* complex subunits in WT and mutant Cyt*b*ΔC13 as determined by complexome profiling. III₂, complex III dimer; III₂/IV₁, III₂/IV₂, supercomplexes containing complex III, and one or two copies of complex IV, respectively. **(B)** Migration profiles of averaged subunits of complexes III and IV from WT and mutant Cyt*b*ΔC13 as determined by complexome profiling from three biological replicates are presented as heatmaps (top) and abundance plots (bottom). III₂, complex III dimer; III₂/IV₁, III₂/IV₂, supercomplexes containing complex III, and one or two copies of complex IV, respectively.

additional insight into complex III assembly and the involvement of the C-terminal region of Cyt*b*.

In the WT strain, all components of intermediate II comigrated at the expected mass of ~90 kD (Fig 4A and Table 1). However, the bulk of Cbp3 and Cbp6, indicative of intermediates 0 and I, was not observed at the predicted masses of 97 and 115 kD, respectively. Instead, they appeared as a peak with a pronounced shoulder at apparent masses that were ~25 kD higher suggesting the presence of additional components. Consistent with its presence in intermediate I but not intermediate 0, Cbp4 peaked at ~140 kD. Because significant amounts of Qcr7 and Qcr8 were also detected in the same mass range of 120–140 kD, we concluded that before Cpb3 and

Cbp6 dissociate to form intermediate II, the two subunits are able to associate and form additional intermediates, here referred to as $0^{7,8}$ and $I^{7,8}$. However, the low relative abundance of Qcr7 and Qcr8, the presence of rather high amounts of Cbp4 in this mass range and a shoulder observed at the lower edge of the main peak indicated the presence of significant amounts of intermediate I.

This notion was corroborated by the pattern of intermediates observed in the migration profiles from mutant Cyt*b*ΔC13. In a range of overlapping peaks at ~80–180 kD, subassemblies at apparent masses indicative of intermediates $0^{7,8}$ and $I^{7,8}$, but also a higher fraction of the canonical intermediates 0 and I were detected (Fig 4B). It should be noted that, in the mutant, Qcr7 was too low in

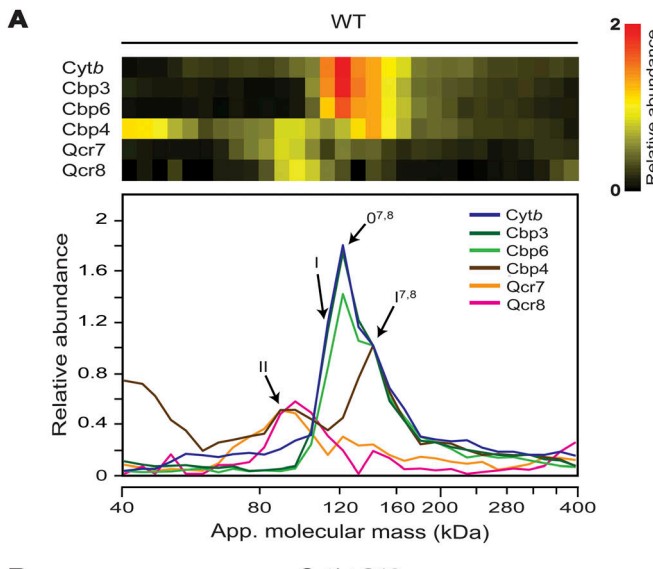

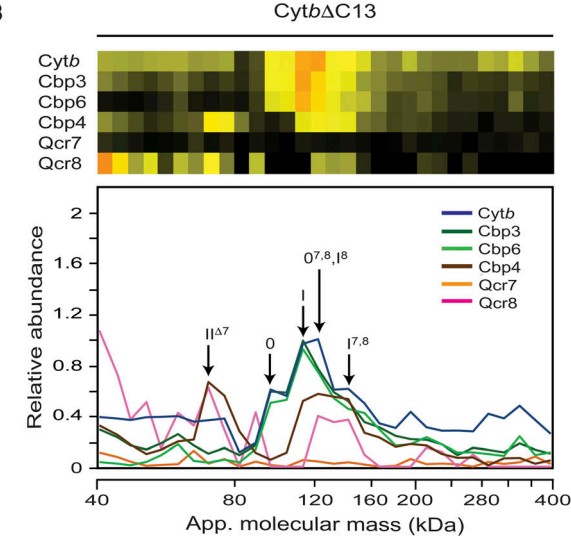

**Figure 4.** **Lack of the Cyt*b* carboxyl terminal region induced changes in accumulation and composition of early-stage assembly intermediates.**
**(A, B)** Abundance and distribution of Cyt*b*, Cbp3, Cbp6, Cbp4, Qcr7, and Qcr8 from WT Cyt*b* (A) and Cyt*b*ΔC13 (B) mitochondria were analyzed by complexome profiling. Heatmaps (top) and migration plots (bottom) show the relative abundance values for the respective proteins. For data visualization and interpretation, the relative abundance of all analyzed proteins in the mass range between 40–400 kD were re-normalized. First, the observed values of Cyt*b*, Cbp3, Cbp6, Cbp4 were normalized to intermediate $I^{7,8}$ at 144 kD and set as 1.0 in the WT strain. Next, the relative abundances of subunits Qcr7 and Qcr8 were normalized to intermediate II and set to the adjusted value of Cbp4 at 91 kD. The profiles of WT and mutant strains were renormalized together. Abbreviations: 0: intermediate 0 (Cyt*b*, Cbp3 and Cbp6); I: intermediate I (Cyt*b*, Cbp3, Cbp6, and Cbp4); II: intermediate II (Cyt*b*, Cbp4, Qcr7, and Qcr8); $0^{7,8}$: intermediate 0 associated with Qcr7/Qcr8; $I^{7,8}$: intermediate I associated with Qcr7/Qcr8; $I^{8}$: intermediate I associated with Qcr8; $II^{Δ7}$: intermediate II lacking Qcr7 (see also Table 1 for extra details). Averaged data from three biological replicates.

abundance to be reliably quantified. This could be because the abundance of all assembly intermediates was markedly diminished by truncation of Cyt*b*. Nevertheless, the presence of Qcr7 in some subassemblies could still be deduced based not only on its migration pattern, but also on the diagnostic apparent masses of

these intermediates established for the WT. The usefulness of this diagnostic mass approach is illustrated by the observation that in the mutant, an intermediate containing Cyt*b* and Cbp4, but not Cbp3/Cbp6, comigrated with Qcr8 at ~70 kD rather than at ~85 kD as predicted for intermediate II indicating the loss of Qcr7. We thus could assign this peak to a sub-assembly $II^{Δ7}$ (Fig 4B). Likewise, the presence of large amounts of Cbp4 and Qcr8 at ~120 kD in the mutant suggested the presence of intermediate $I^{8}$. The apparent molecular masses of the assigned intermediates matched remarkably well with the predicted values deviating just ≤5 kD (Table 1).

Steady state levels of the intermediates were overall reduced in the mutant. Moreover, the observation that in the Cyt*b*ΔC13 mutant, canonical intermediates 0 and I were present in markedly higher amounts than the ones already associated with Qcr7 and Qcr8, indicating that the C-terminal region of Cyt*b* must be involved in this interaction. Occurrence of intermediates $I^{8}$ and $II^{Δ7}$, still containing Qcr8 but lacking the other subunit, suggested the incomplete binding of Qcr7 to the truncated Cyt*b* as the primary cause for a weaker association of both proteins.

### Truncation of the Cyt*b* C-terminal region does not prevent the association of cytochrome *b* with Cbp3

Cbp3 has a direct, physical interaction with Cyt*b*, and some Cbp3-specific residues involved in this interaction have been identified, that is, Gln-183, Lys-185, Asp-188, Glu-195, and Lys-215 (Ndi et al, 2019). These residues, located in the C-terminal half of Cbp3, form a cleft where Cyt*b* may be recruited. Deletion of the Cyt*b* C-terminal region did not abrogate the Cyt*b*–Cbp3 interaction, but reduced accumulation of intermediate I, increased the abundance of intermediate 0, prevented the formation of intermediate II and induced the appearance of aberrant subassemblies, for example, $II^{Δ7}$. A significant accumulation of free Cbp3/Cbp6 complex was also observed in the mutant. These results suggested a lower affinity of Cbp3/Cbp6 with Cyt*b* upon its C-terminal truncation.

To investigate if some of the previously identified interacting residues of Cbp3 could no longer interact with Cyt*b*ΔC13, we analyzed photo-crosslinked products of Cbp3 and Cyt*b*ΔC13 using the nonnatural, photo-activable amino acid *p*-aminobenzoyl-phenylalanine (pBpa). Plasmids coding for Cbp3-His$_7$ were mutagenized to introduce an amber stop codon at the desired positions, Lys-185, Asp-188, and Lys-215. Δ*cbp3* cells carrying one of these plasmids were incubated with pBpa in the dark. Then, mitochondrial translation products were labeled with ($^{35}$S)-methionine in the presence of cycloheximide to detect Cyt*b* in cross-linked products. The cells were further incubated under UV light to photo-crosslink Cbp3-His$_7$. Cbp3-His$_7$ cross-linked products were purified using Ni-NTA beads. Samples were analyzed by SDS–PAGE and autoradiography, and by Western blot using antibodies against Cbp3. The Cyt*b*-Cbp3-His$_7$ cross-linking product was previously identified at ~64 kD (Ndi et al, 2019). We also detected this product by looking at ($^{35}$S)-methionine-labeled products (Fig S4A) and by Western blot against Cbp3 (Fig S4B). Similar crosslinked products were observed for Cyt*b* and for Cyt*b*ΔC13, indicating that, at least Lys-185, Asp-188, and Lys-215 from Cbp3 are still interacting with Cyt*b*, even when the

**Table 1.  Composition of $bc_1$ complex assembly intermediates observed in WT Cyt$b$ and Cyt$b\Delta$C13 mitochondria by complexome profiling.**

|  | Intermediate | App. $M_r$ | Calc. $M_r$ | $\Delta M_r$ | Cyt$b$ | Cbp3 | Cbp6 | Cbp4 | Qcr7 | Qcr8 |
|---|---|---|---|---|---|---|---|---|---|---|
|  |  | kD | kD | kD |  |  |  |  |  |  |
| Wild-type | II | 91 | 87 | 4 | X |  |  | X | X | X |
|  | 0[7,8] | 124 | 123 | 1 | X | X | X |  | X | X |
|  | I | 115 | 115 | 0 | X | X | X | X |  |  |
|  | I[7,8] | 144 | 141 | 3 | X | X | X | X | X | X |
| Cyt$b\Delta$C13 | II$^{\Delta 7}$ | 72 | 71 | 1 | X |  |  | X |  | X |
|  | 0 | 98 | 95 | 3 | X | X | X |  |  |  |
|  | 0[7,8] | 124 | 121 | 3 | X | X | X |  | X | X |
|  | I | 115 | 113 | 2 | X | X | X | X |  |  |
|  | I[8] | 124 | 124 | 0 | X | X | X | X |  | X |
|  | I[7,8] | 144 | 139 | 5 | X | X | X | X | X | X |

The columns show the apparent molecular masses (App. $M_r$) compared with the theoretical calculated masses (Calc. $M_r$) of the different subunits and chaperones present in early-stage $bc_1$ complex assembly intermediates. The difference between apparent and calculated molecular masses is shown in column $\Delta M_r$. Presence of each subunit or chaperone as component of a given intermediate is indicated with an "X". Comparison of experimental apparent molecular mass (App. $M_r$) and calculated molecular mass (Calc. $M_r$), and the difference between these two values ($\Delta M_r$) are indicated.

C-terminal region is missing. Accordingly, the suggested weaker interaction of Cbp3 with Cyt$b$ must involve other contact sites.

## Discussion

Cytochrome $b$ is the only subunit of complex III that is encoded in the mitochondrial genome, and it is a key component for its catalysis and assembly. Synthesis of Cyt$b$ depends on chaperones Cbp3/Cbp6 (Gruschke et al, 2011) and is highly coupled not only to its hemylation states, but also to assembly of the complex (Gruschke et al, 2012; Hildenbeutel et al, 2014). In the present work, we demonstrated that the carboxyl terminal region of Cyt$b$ has an essential role in these processes.

We sought to investigate the function of the Cyt$b$ C-terminal region because (*i*) it is a hydrophilic region of the protein facing the mitochondrial matrix, potentially placing it close to chaperones and translational activators at early assembly stages; (*ii*) subunit Qcr7 interacts with the Cyt$b$ C-terminal region, and absence of this subunit triggers the *COB* mRNA translational repression by an assembly feedback loop (Gruschke et al, 2012; García-Guerrero et al, 2018); and (*iii*) once subunit Qcr7 fully associates, Cbp3/Cbp6 release from intermediate I, suggesting that Cbp3/Cbp6 and Qcr7 should share common sites of interaction on Cyt$b$.

As the nascent Cyt$b$ polypeptide emerges from the mitoribosome, it associates with Cbp3 and Cbp6 to form intermediate 0 and subsequently, intermediate I. The current yeast $bc_1$ complex assembly model proposes that incorporation of subunits Qcr7 and Qcr8 triggers the release of Cbp3/Cbp6 to form intermediate II (Gruschke et al, 2012); therefore, Cbp3/Cbp6, which are rate limiting for Cyt$b$ synthesis, are available to activate more *COB* mRNA translation. If either Qcr7 binding or Cyt$b$ hemylation do not take place, Cbp3/Cbp6 are then sequestered in intermediates 0 and I and translation of the *COB* mRNA decreases through an assembly-feedback regulatory

mechanism (Gruschke et al, 2012; Hildenbeutel et al, 2014). Remarkably, deletion of the Cyt$b$ C-terminal region abrogates this assembly-feedback regulatory loop, because a Cyt$b\Delta$C13/$\Delta$qcr7 double mutant showed normal levels of Cyt$b$ synthesis. Our data indicate that Cbp3/Cbp6-containing subassemblies are observed in the absence of the Cyt$b$ C-terminal region, and that significant amounts of free Cbp3 and Cbp6 and Cpb3/Cbp6 heterodimer accumulate. This could restore *COB* mRNA translation even in the absence of Qcr7. We thus propose that the absence of the Cyt$b$ C-terminal region leads to a weaker interaction between Cbp3/Cbp6 and Cyt$b$, which makes the chaperones more prone for self-release. It remains elusive how different regions of the Cyt$b$ protein (like soluble loops or transmembrane specific residues) could exert regulatory roles on *COB* mRNA translation. Interestingly, mutation of the catalytic heme $b$-binding sites on Cyt$b$ decreased synthesis (Hildenbeutel et al, 2014), whereas deletion of the last 8–13 residues from the Cyt$b$ C-terminus released translational regulation.

Even though absence of Cyt$b$ C-terminal region decreased the abundance of the predominant early-assembly intermediate I, it did not prevent its formation. It was previously observed that Cyt$b$, in association with Cbp3/Cbp6, receives the heme $b_L$, and incorporation of Cbp4 stabilizes this hemylation, and facilitating the addition of heme $b_H$ to transit into formation of intermediate II (Hildenbeutel et al, 2014). It is unclear whether the identified early subassemblies in the Cyt$b\Delta$C13 mutant contain Cyt$b$ with heme(s) $b$ or not. High accumulation of the canonical heme $b_H$-less intermediate 0 was consistently observed in the mutant profiles. Three subassemblies containing Cbp4 were also detected in Cyt$b\Delta$C13. At this point, it is not known if the missing Cyt$b$ C-terminal region might block heme $b_L$ insertion even if Cbp3/Cbp6 are associated. In addition, canonical intermediate II was not detected in the Cyt$b\Delta$C13 mutant. An aberrant subassembly containing Cyt$b$, Cbp4, and Qcr8 (II$^{\Delta 7}$) accumulated in its place. Lack of Qcr7 in this

subassembly is not surprising because its major interaction site, the Cyt*b* C-terminus, is absent. This may explain the poor stability of this subunit in this complex. All these results suggest that the Cyt*b* C-terminal region is crucial to regulate the Cbp4 interaction with intermediate I and stability of Qcr7 within *bc₁* complex.

Interestingly, our complexome profiling analysis revealed additional subassemblies containing the same components as intermediates 0 and I but migrating at slightly higher molecular masses. These subassemblies were consistently found in all analyzed biological replicates from WT and Cyt*b*ΔC13 strains. To further investigate which additional interactor(s) caused the observed mass shift, we carefully checked the list of identified proteins of ~15–25 kD that comigrated with Cyt*b*, Cbp3, Cbp6, and Cbp4 in the mass range of 50–200 kD. Although we found several proteins with similar migration patterns, the profiles of Qcr7 and Qcr8 showed the best match with Cyt*b* and other chaperones. Moreover, their molecular masses fitted well to the observed differences in the apparent molecular masses. Hence, we propose that the two subunits can bind already to intermediates 0 and I. Such subassemblies, that is, $0^{7,8}$ and $I^{7,8}$ (Fig 4 and Table 1) might denote parallel assembly pathways occurring during Cyt*b* folding and binding/releasing of subunits/chaperones ultimately converging in the formation of stable intermediates I and II. Incorporation and settled interaction of such subunits is what could promote the final releasing of Cbp3/Cbp6. Nevertheless, it is not straightforward to understand why Qcr7/Qcr8 bind to intermediate 0. We speculate that there are interaction sites in Cyt*b* that might not be occupied by Cbp3/Cbp6 where Qcr7/Qcr8 can interact early and wait until Cbp4 binds to promote conformational changes that allow their final mode of interaction. Further structure and mechanistic research will be required to validate the existence of these subassemblies, and the specific molecular roles of all early assembly-involved complex III chaperones.

Chaperone Cbp3 interacts physically and directly with Cyt*b* (Gruschke et al, 2012; Ndi et al, 2019). Lys-185, Asp-188, and Lys-215 are some of the amino acids from Cbp3 that directly interact with Cyt*b* and localize at an extended surface area on the chaperone (Ndi et al, 2019). All three Cbp3 amino acids are still interacting with mutant Cyt*b*, indicating that the Cbp3–Cyt*b* interaction takes place probably with one of the three matrix-side soluble Cyt*b* loops. Even though the Cyt*b* carboxyl terminal region is not absolutely required for Cbp3 association, this region of the protein seems to regulate how tightly Cbp3/Cbp6 associates with cytochrome *b*. This function might be achieved by Cyt*b* C-terminal region conformational changes throughout the progression of this subunit through the assembly intermediates.

The Cyt*b* C-terminal region could also regulate how Cbp4 incorporates and executes its role(s) in the assembly complex. This could be a direct or indirect role through the correct positioning of Cbp3/Cbp6 and Qcr7/Qcr8 in intermediates 0 and $0^{7,8}$. This is of particular importance because truncation of Cyt*b* did not prevent the formation of Cbp4-containing subcomplexes ($I^8$, $I^{7,8}$, and $II^{Δ7}$), that seem to be not functional, nonetheless. If hemylation proceeds correctly, Cbp3/Cbp6 are then released, which may be promoted not only by association of subunits Qcr7/Qcr8, but also by a new conformation of the Cyt*b* C-terminal region enabling intermediate I to transit into intermediate II. Indeed, stability of intermediate II

seems to depend on the presence of the Cyt*b* C-terminal region, as we were unable to detect this intermediate in the mutant lacking this region.

In summary, our data demonstrate that the Cyt*b* C-terminal region has an essential role in regulation of the assembly-feedback loop of *COB* mRNA translation, formation of early-assembly intermediates and, thereby, assembly of complex III. Based on previous work and the results presented here, we propose an updated model in which, as Cyt*b* emerges from the mitoribosome, it first interacts with Cbp3 and Cbp6. These interactions are promoted by a specific conformation of the Cyt*b* carboxyl terminal region (Fig 5) to assemble into intermediate 0. Correct formation of this intermediate might trigger the addition of heme $b_L$. Cbp4 is recruited to stabilize heme $b_L$, and heme $b_H$ is added to conform intermediate I (Hildenbeutel et al, 2014). Interestingly, our data also suggest an alternative early-assembly route where intermediate 0 already binds Qcr7/Qcr8. Then, Cbp4 attaches and may help stabilize Cyt*b* folding and proper interactions of Qcr7/Qcr8, which makes Cbp3/Cbp6 lose affinity for Cyt*b* and detach to form the intermediate II.

Finally, we would like to emphasize that the carboxyl terminal region of other mitochondrially encoded proteins is important for the biogenesis of their respective oxidative phosphorylation complexes. As the Cyt*b* C-terminal region is essential at different steps of complex III biogenesis, it has also been shown for the carboxyl terminal region of Cox1, the largest subunit of cytochrome *c* oxidase (Shingú-Vázquez et al, 2010; García-Villegas et al, 2017). In this case, deletion of the last 15 residues of the Cox1 C-terminal region abrogates the assembly-feedback regulation of Cox1 synthesis with a cascade of defects in cytochrome *c* oxidase biogenesis (García-Villegas et al, 2017). Moreover, biogenesis of complex II, which is only formed by nuclear-encoded subunits, is also regulated by the C-terminal region of subunits Sdh1 and Sdh4 (Oyedotun & Lemire, 1997; Kim et al, 2012). This mechanism of regulation is not exclusively found in the mitochondria because similar findings were reported for chloroplast-encoded mRNAs (Choquet & Wollman, 2002). All these studies highlight the intricate regulation of translation and assembly involving a small soluble region of, in this case, mitochondrial proteins. Such domains seem to play a special role in maintaining the concerted formation and coordinated assembly of nuclear and organellar-encoded subunits, particularly at early-assembly stages of the biogenesis of multi-protein complexes of dual genomic origin.

# Materials and Methods

### Strains, plasmids, medium, and genetic methods

*Saccharomyces cerevisiae* strains used are congenic or isogenic to *BY4742* and *D273-10b* (listed in the Table S1). Genetic methods and medium were previously described (Burke et al, 2000; Dunham et al, 2015). Cells were grown in complete fermentative medium (1% yeast extract, 2% Bacto-peptone, and 2% glucose or 2% galactose), synthetic complete medium (0.67% yeast nitrogen base, 2% glucose, and without uracil or the amino acids to select), respiratory medium (1% yeast extract, 2% Bacto-peptone, 3% ethanol and 3% glycerol).

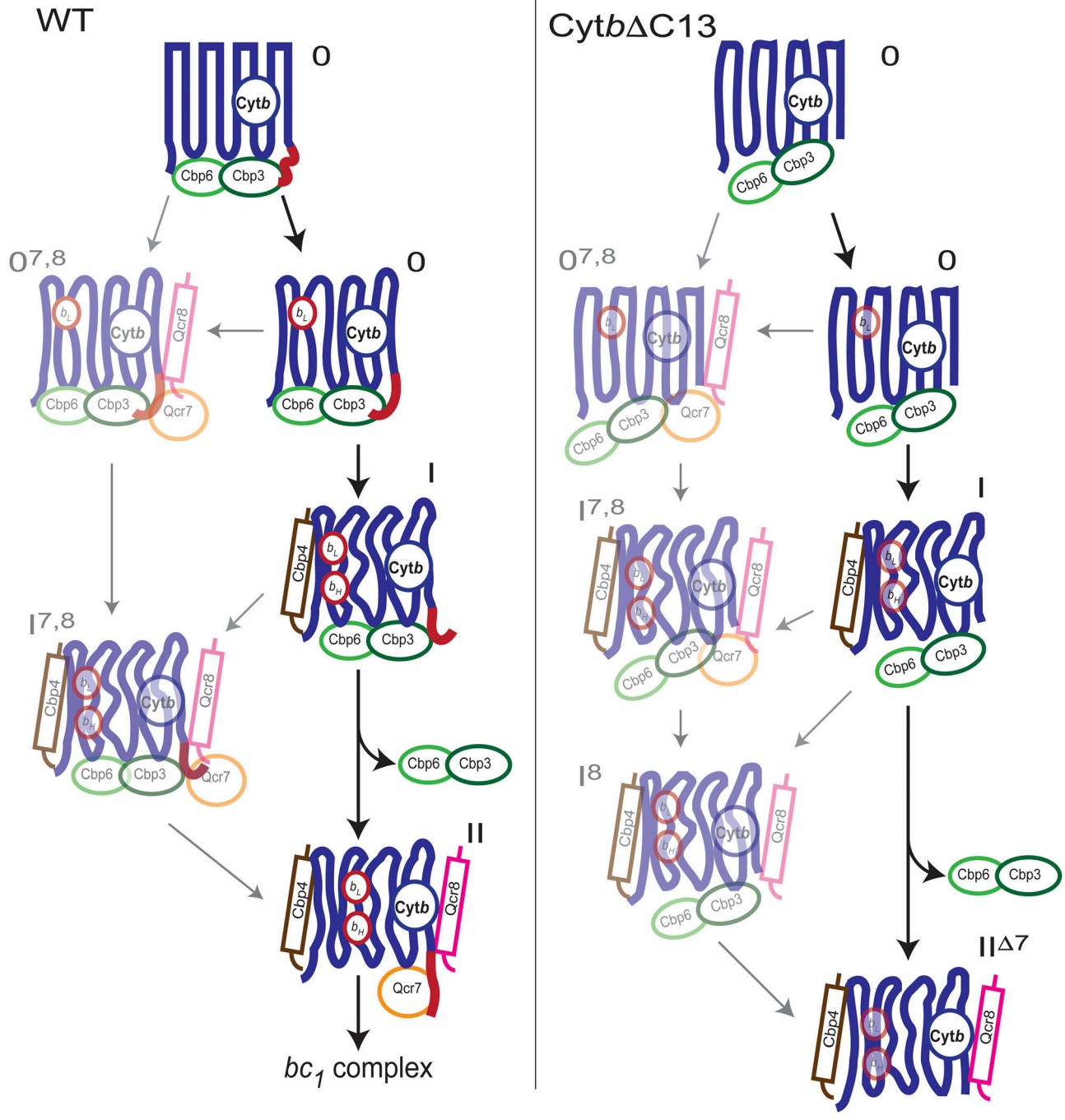

**Figure 5. Schematic model for participation of the Cyt*b* carboxyl terminal region on regulation of Cyt*b* synthesis and *bc₁* complex assembly.**
After translational activation of the *COB* mRNA, the Cyt*b* C-terminal region, exposed to the mitochondrial matrix, changes conformation throughout early assembly intermediates 0, I, II, and in low abundance $0^{7,8}$ and $I^{7,8}$. This region of the protein allows correct association with Cbp3/Cbp6 for Cyt*b* synthesis regulation. The Cyt*b* C-terminal region changes until it reaches its final conformation once *bc₁* complex is fully assembled. If the Cyt*b* C-terminal region is missing, then there is an overall decreased accumulation of intermediates 0, I, $0^{7,8}$, and $I^{7,8}$; intermediate II is lost and instead, intermediates $II^{Δ7}$ and $I^8$ appear. Upon truncation of Cyt*b* C-terminus, Cbp3/Cbp6 binding is inefficient, and free Cbp3/Cbp6 heterodimers are accumulated. Consequently, the assembly feedback regulation of Cyt*b* synthesis is lost, so now, Cyt*b* is efficiently translated even if the Qcr7 subunit is missing. Because canonical intermediate II is not formed in the mutant, the assembly of the *bc₁* complex is abrogated. Low abundant, not previously described intermediates $0^{7,8}$, $I^{7,8}$, and $I^8$ are shown as transparent images.

Deletion of nuclear genes was made by PCR amplification of cassettes where the *orf* of interest was replaced with *LEU2*, *HIS3*, *URA3* or resistance to G418 (*KanMX4*). This DNA was used to transform yeast by the lithium acetate method. *CBP3-HA* was amplified by PCR and cloned on the yeast expression plasmid pRS315 to create plasmid pDM8.

## Mitochondrial transformation

Mutant version of *CYTB* coding for proteins lacking the last 8 (Cyt*b*ΔC8) or 13 residues (Cyt*b*ΔC13) were made by fusion PCR and subcloned in pDFM2 plasmid (containing the *CYTB* sequences cloned in pBluescript KS). Plasmid and PCR products were digested with *Sph1* and *Kpn1* and ligated to obtain plasmids pDFM4 (Cyt*b*ΔC8) and pDFM6 (Cyt*b*ΔC13). The plasmids were transformed into NAB69 rho0 strain by high-velocity microprojectile bombardment (Biolistic PDS-1000/He Particle Delivery System; BIO-RAD) (Bonnefoy & Fox, 2007) to create the Rho-synthetic strains DFM7 and DFM8 containing *CYTBΔC13* and *CYTBΔC8,* respectively. *CYTB* mutants were inserted into DFM2 (strain derived from BY4742, that has intronless *COX1* and the construct *cytbΔ::ARG8$^m$*) mitochondria by cytoduction to create DFM9 and DFM10. The rho⁻ synthetic *CYTBΔC13* (DFM7) was inserted into DFM16 (strain derived from D273-10b, that has intronless *COX1* and the construct *cytbΔ::ARG8$^m$*) mitochondria by cytoduction to create DFM24. The strains were tested with M17-162 4D that contains the *CYTB2* allele (Bonjardim et al, 1996). The mutants were corroborated by PCR and sequencing.

## Analysis of mitochondrial proteins

Cell cultures were grown in complete fermentative medium with 2% galactose (YPGal) until the late exponential phase. Mitochondria were isolated breaking the cell with Zimolyase 20T and with glass beads (Diekert et al, 2001). The mitochondrial proteins were measuring by Lowry method (Markwell et al, 1978) and separated for SDS–PAGE (12% and 16%), transferred to PVDF membrane, and inmunodecorated with antibodies against anti-HA (12013819001; Roche Applied Science), anti- Cyt*b* (García-Guerrero et al, 2018), anti-Rip1 (Rosemary Stuart), anti-Cbp3, anti-Mrp20 (Thomas D Fox), anti-Citrate synthase (Thomas D Fox), anti-mouse HRP (SC-2005; Santa Cruz Biotechnology), and anti-rabbit HRP (111035144; Jackson ImmunoResearch) secondary antibodies. For immunoprecipitation, mitochondria (500 *μg*) were solubilized with 1% digitonin (wt/vol) and separated by centrifugation, the supernatant was incubated with Protein A-Sepharose beads coupled with anti-HA antibody. Total (10%) and immunoprecipitated fraction (90%) were collected and separated for SDS–PAGE.

## Differential spectra analysis

For hemes spectrophotometric analysis in isolated mitochondria, samples (1.5 mg protein) were resuspended in phosphate buffer pH 6.6. Differential spectra were determined by the subtraction of dithionite-reduced minus ferricyanide-oxidized state in a DW2 Aminco UV-Visible spectrophotometer modernized by OLIS Inc. (On-Line Instrumental Systems, Inc.).

## Synthesis of mitochondrial proteins

In vivo labeling of cells in the presence of ($^{35}$S)-methionine (7 *μCi*) and cycloheximide was performed as previously described (Perez-Martinez et al, 2003). After 15 min of pulse labeling, cells were chilled on ice/water and disrupted by vortexing with glass beads to obtain the mitochondria by centrifugation. Mitochondrial proteins were separated on a 16% polyacrylamide gel, transferred to a PVDF membrane, and analyzed with a Typhoon 8600 Phosphorimager (GE Healthcare). In organello translation assays were carried out incubating isolated mitochondria (500 *μg*) with [$^{35}$S]-methionine (20 *μCi*) for 20 min, as previously described (Westermann et al, 2001).

## Mitoribosome separation on a sucrose cushion

Purified mitochondria (300 *μg*) were lysed with 400 *μl* of lysis buffer (1% digitonin, 50 mM KCl, 0.5 mM MgCl$_2$, 20 mM Hepes/KOH, pH 7.4, 1× Complete Protease Inhibitor mix [Roche]) for 30 min on ice. The samples were centrifuged at 16,100*g* 15 min at 4°C. 200 *μl* of the supernatant were precipitated with 12% TCA (total fraction) and the rest was loaded on 50 *μl* of a cushion (1.2 M sucrose, 20 mM Hepes/KOH, pH 7.4) and ultracentrifuged for 105 min at 200,000*g* at 4°C in a TLA100 rotor (Beckman Coulter). The supernatant and the pellet fraction (mitoribosomes) were resuspended with lysis buffer and precipitated with 12% TCA. The fractions were analyzed by SDS–PAGE and Western blot.

## Blue-native electrophoresis

BN-PAGE was performed as previously described Wittig et al (2006). Samples (100 *μg*) of mitochondrial protein were solubilized with 20 *μl* of 50 mM Bis-Tris pH 7.0, 750 mM aminocaproic acid, and either 1–2% digitonin or 1% n-dodecyl *β*-D-maltoside (DDM) on ice. Mitochondrial extracts were cleared at 30,000 x g for 30 min, and the supernatants were mixed with 2.5 *μl* of Coomassie solution (5% Coomassie, 750 mM aminocaproic acid, 50 mM Bis-Tris). Extracts were loaded on a 5–12% polyacrylamide gel and transferred to a PVDF membrane. Proteins were detected by immunoblotting with the indicated antibodies. In-gel CIV activity was performed after BN-PAGE using 0.04% diaminobenzidine (Sigma-Aldrich) and 0.02% of horse heart cytochrome *c* (Sigma-Aldrich) in phosphate buffer pH 7.4 (Wittig et al, 2007). For second dimension analysis, one BN gel lane was cut and separated on 12% SDS–PAGE.

## Isolation of yeast mitochondria membranes for complexome profiling and enzyme activity assays

Small-scale isolation of unsealed mitochondrial membranes was done as follows. Freshly harvested yeast cells (wet weight ~10 g) were resuspended in 10 ml mitobuffer (600 mM D-mannitol, 2 mM EGTA, 2 mM PMSF, 10 mM Tris–HCl pH 6.8). Yeast cells were cracked with glass beads (10 g) by high-speed vortexing; 10 × 1 min pulses with 1-min resting intervals on ice. The unbroken cells, nuclei and glassbeads were pelleted by centrifugation at 3,300*g* for 20 min at 4°C. The supernatant was decanted into a fresh tube. The remaining pellet was rinsed again with fresh 10 ml mitobuffer and centrifuged as in the previous step. The supernatant was recovered and pooled with the first one followed by high-speed centrifugation at 40,000*g* for 60 min at 4°C. The resulting pellets of crude mitochondrial membranes were resuspended in 500 *μl* mitobuffer, carefully homogenized, aliquoted, shock-frozen in liquid nitrogen, and stored at –70°C. The aliquots were thawed on ice before use

and respective protein concentrations were determined using the Bio-Rad DC protein assay.

## Complexome profiling

Isolated mitochondrial membranes (100 µg) were solubilized with digitonin (3 mg/mg protein) and separated by BN-PAGE using a 4–16% polyacrylamide gradient gel as described previously (Wittig et al, 2006). After the run, proteins were fixed overnight in 50% methanol, 10% acetic acid, 100 mM ammonium acetate. Gels were further stained with 0.025% Coomassie blue G-250 (Serva G) in 10% acetic acid for 45 min, distained with 10% acetic acid, and maintained in deionized water until maximal size recovery. After scanning gels, the real size images were used as templates for the cutting procedure.

Proteins were identified by tandem mass spectrometry (MS/MS) after in-gel tryptic digestion following the procedure described in (Heide et al, 2012) with a few modifications. Briefly, entire gel lanes were cut in 60 uniform slices starting at the bottom. The slices were chopped and transferred into 96-well filter plates (MABVN1250; Millipore) adapted manually to 96-well plates (MaxiSorpTM Nunc) as waste collectors. Gel pieces were incubated in 50% methanol, 50 mM ammonium bicarbonate (ABC) under moderate shaking; the solution was refreshed until all the blue dye was removed. In all cases, removal of excess solution was done by centrifugation (1,000g, 20 s). Then, gel pieces were reduced with 10 mM dithiothreitol in 50 mM ABC for 1 h. After removing the excess solution, 30 mM chloroacetamide in 50 mM ABC was added to each well, incubated in the dark for 45 min, and removed. A 15-min incubation step with 50% methanol, 50 mM ABC was performed to dehydrate gel pieces. Excess solution was removed, and gel pieces were dried for 30–45 min at room temperature. To digest proteins, 20 µl of 5 ng·µl$^{-1}$ trypsin (sequencing grade, Promega) in 50 mM ABC, 1 mM CaCl$_2$ were added to each well and incubated for 20 min at 4°C. Later, gel pieces were totally covered by adding 50 µl of 50 mM ABC followed by an overnight incubation at 37°C. The next day, the diffused peptides containing supernatants were collected by centrifugation (1,000g, 60 s) into clean 96-well PCR plates (Axygen). The gel pieces were lastly incubated with 50 µl of 30% acetonitrile (ACN), 3% formic acid (FA) for ~20 min before elution of the remaining peptides on the previous eluates by centrifugation. The peptides were dried in a SpeedVac Concentrator Plus (Eppendorf) for 3 h, resuspended in 30 µl of 5% ACN, 0.5% FA, and stored at −20°C until MS analysis.

After thawing and gentle shaking of the resuspended peptides for ~20 min, each fraction was loaded and separated by reverse-phase liquid chromatography (LC) and analyzed by tandem MS/MS in a Q Exactive orbitrap mass spectrometer equipped with a nano-flow ultra-HPLC system (Easy nLC-1000; Thermo Fisher Scientific). In short, peptides were separated using 100 µm ID × 15 cm length PicoTip EMITTER columns (New Objective) filled with ReproSil-Pur C18-AQ reverse-phase beads (3 µm, 120 Å) (Dr. Maisch GmbH) using linear gradients of 5–35% ACN, 0.1% FA (30 min) at a flow rate of 300 nl min$^{-1}$, followed by 35–80% ACN, 0.1% FA (5 min) at 600 nl min$^{-1}$, and a final column wash with 80% ACN (5 min) at 600 nl min$^{-1}$. All settings for the mass spectrometer operation were the same as detailed in (Guerrero-Castillo et al, 2017). The 60 fractions of each

yeast strain were analyzed thrice by LC–MS/MS (three biological replicates).

MS raw data files from all individual fractions were analyzed using MaxQuant (v1.5.0.25) against the *S. cerevisiae* reference proteome entries retrieved from Uniprot (UP000002311, 26.03.2022). Settings applied: Trypsin/P, as protease, N-terminal acetylation, deamidation (NQ), and methionine oxidation (1 and 2 sites) as variable modifications; cysteine carbamidomethylation as fixed modification; two trypsin missed cleavages; matching between runs, 2 min matching time window; six residues as minimal peptide length; common contaminants included, I = L. All other parameters were kept as default. Individual protein abundances were determined by label-free quantification using the obtained intensity-based absolute quantification values. To account for gel slicing, protein loading, and MS sensitivity variations, complexome profiles were aligned and normalized against the list of identified yeast proteins annotated as mitochondrial in UniProt using the tool COPAL (Van Strien et al, 2019). For each protein group, migration profiles were generated and normalized to the maximal abundance through all fractions (relative abundance). Hierarchical clustering of the migration patterns of all identified proteins was applied using an average linkage algorithm with centered Pearson correlation distance measures. Resulting complexome profiles consisting of a list of proteins arranged according to the similarity of their migration patterns in BN-PAGE were visualized as heatmaps representing the normalized abundance in each gel slice by a three-color gradient (black/yellow/red) and processed in Microsoft Excel 2019 for analysis. Data from three biological replicates were averaged to generate the final complexome profiles. For proteins Qcr7 and Qcr8, low-score peptides that were not found in all replicates were manually excluded from the analysis. The mass calibration for the BN gel was performed using the apparent molecular masses of well-known membrane and soluble proteins from *S. cerevisiae* (see below for data availability).

## Enzyme activity assays

Activity measurements were carried out for succinate:decylubiquinone reductase (Complex II), decylubiquinol:cytochrome *c* reductase (complex III), combined complexes II+III, and cytochrome *c* oxidase (complex IV) by spectrophotometric measurements using a SpectraMax ABS Plus microplate reader (Molecular Devices). Unsealed mitochondrial membranes were thawed on ice and diluted to 100 µg/ml in 25 mM potassium phosphate buffer (pH 7.5) with 2 mM PMSF. In all cases, activity assays were performed at 25°C, recorded for 3 min using 20 µg/ml mitochondrial membranes. Complex II activities were measured in 25 mM potassium phosphate buffer (pH 7.5), 10 mM succinate, 1 mg/ml BSA, 2 µM antimycin A, 500 µM sodium cyanide, and 80 µM 2,6-dichlorophenolindo-phenol (DCPIP). The reactions were initiated with the addition of 70 µM decylubiquinone (DBQ) and resultant reduction of DCPIP was followed at 600 nm ($\varepsilon_{600nm}$ = 19.1 mM$^{-1}$·cm$^{-1}$). Furthermore, 10 mM malonate was added to inhibit the reaction and the residual activity was subtracted accordingly. Complex III activities were measured in 25 mM potassium phosphate buffer (pH 7.5), 0.1 mM EDTA, 500 µM NaCN, 100 µM decylubiquinol (DBQH$_2$). The reactions were initiated with the addition of 75 µM oxidized

cytochrome $c$ and its reduction was followed at 550 nm ($\varepsilon_{550nm}$ = 18.5 mM$^{-1}\cdot$cm$^{-1}$). 2 $\mu$M antimycin A was added to inhibit the reaction and the residual activity was later subtracted. Similarly, Complex II+III activities were measured in 25 mM potassium phosphate buffer (pH 7.5), 1 mg/ml BSA, 10 mM succinate, 500 $\mu$M NaCN. The reactions were initiated with the addition of 75 $\mu$M oxidized cytochrome $c$ and its reduction was followed at 550 nm. Residual activities in the presence of 10 mM malonate were subtracted from the respective measurements. Complex IV activities were measured by after the oxidation of cytochrome $c$ at 550 nm in 25 mM potassium phosphate buffer (pH 7.0) with 2 $\mu$M antimycin A. The reactions were initiated with the addition of 50 $\mu$M-reduced cytochrome $c$. 1 mM NaCN was added to inhibit the reaction and the residual activities were subtracted. To account for variation between preparations, all data were corrected against complex II activities. The details for normalization of the respective activities are shown in the legend of Fig S3.

### Photo-cross linking of Cbp3

$\Delta cbp3$ cells that contained EcYRS-Bpa (*E. coli* tyrosyl-tRNA synthetase) and tRNA (*E. coli* tyrosyl tRNA$_{CUA}$) integrated in the yeast nuclear genome; were transformed with a plasmid containing *CBP3-HIS$_7$* with amber stop codons at the desired positions. Cells were grown o/n in 50 ml of minimal medium (SGal-Leu) with 250 $\mu$l of 200 mM pBpa in darkness and harvested in logarithmic growth phase. Cells were radiolabeled with [$^{35}$S]-methionine in the presence of cycloheximide for in vivo translation for 40 min, as described above. The cells were resuspended in SGal-Leu and transferred to a 12 well-plate to irradiate with UV light at 350 nm for 45 min. Cells were then treated with 0.1 M NaOH for 5 min at room temperature and lysate with 140 $\mu$l of 4% SDS, 100 mM DTT for 5 min at 95°C. The cross-linked products were purified on Ni-NTA beads (Qiagen) as previously described (Ndi et al, 2019). Proteins were separated by SDS–PAGE 16%, transferred to PVDF membrane, revealed by autoradiography, and immunodecorated with antibodies against Cbp3.

## Data Availability

The complexome profiling dataset was deposited in the ComplexomE profiling DAta Resource (CEDAR) (Van Strien et al, 2019) website (https://www3.cmbi.umcn.nl/cedar/browse/experiments/CRX37) for free public access with the accession code CRX37. The mass calibration curves are available in the submitted processed dataset file (tab: Mass calibration). Raw data and other datasets generated and/or analyzed during this study are available from the corresponding authors upon appropriate request.

## Supplementary Information

## Acknowledgements

We thank Thomas D Fox, Rosemary Stuart for the gift of antisera; Thomas D Fox, Gabriel del Río-Guerra and Teresa Lara-Ortiz for the gift of yeast strains. We thank Martin Ott and Andreas Carlström (Stockholm University) for advice on the experiment of Fig S4. We thank the technical support provided by Tecilli Cabellos-Avelar and Enrique Chávez. We thank Ulrik Pedroza-Ávila for careful reading of the manuscript. This work was supported by research grants from Consejo Nacional de Ciencia y Tecnología (CONACyT) (47514 to X Pérez-Martínez), and fellowship (777444 to D Flores-Mireles; 255917 to AE García-Guerrero); and Programa de Apoyo a Proyectos de Investigación e Innovación Tecnológica, (PAPIIT), UNAM (IN202720 and IN223623 to X Pérez-Martínez). A Cabrera-Orefice, M Lutikurti and U Brandt are supported by the Netherlands Organization for Health Research and Development (ZonMW TOP-Grant 91217009). This manuscript is part of the doctoral dissertation of Daniel Flores-Mireles from the Programa de Maestría y Doctorado en Ciencias Bioquímicas, Universidad Nacional Autónoma de México.

### Author Contributions

D Flores-Mireles: investigation and writing—original draft.
Y Camacho-Villasana: investigation and methodology.
M Lutikurti: formal analysis and investigation.
AE García-Guerrero: supervision and investigation.
G Lozano-Rosas: resources.
V Chagoya: resources and software.
EB Gutiérrez-Cirlos: resources and supervision.
U Brandt: resources and formal analysis.
A Cabrera-Orefice: formal analysis, supervision, investigation, and writing—original draft.
X Pérez-Martínez: conceptualization, supervision, funding acquisition, and writing—original draft, review, and editing.

### Conflict of Interest Statement

The authors declare that they have no conflict of interest.

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
