## [Reviewer comments · Life Science Alliance]

Life Science Alliance

The cytochrome b carboxyl-terminal region is necessary for mitochondrial Complex III assembly

Daniel Flores-Mireles, Yolanda Camacho-Villasana, Madhurya Lutikurti, Aldo García-Guerrero, Guadalupe Lozano-Rosas, Victoria Chagoya de Sanchez, Emma Berta Gutiérrez-Cirlos, Ulrich Brandt, Alfredo Cabrera-Orefice, and Xochitl Pérez-Martínez
DOI: <https://doi.org/10.26508/lsa.202201858>

Corresponding author(s): Xochitl Pérez-Martínez, Universidad Nacional Autónoma de México

Review Timeline:

Submission Date:	2022-11-27
Editorial Decision:	2023-01-23
Revision Received:	2023-03-15
Editorial Decision:	2023-04-03
Revision Received:	2023-04-09
Accepted:	2023-04-11

Scientific Editor: Novella Guidi

Transaction Report:

January 23, 2023

Re: Life Science Alliance manuscript #LSA-2022-01858-T

Dr. Xochitl Pérez-Martínez
Instituto de Fisiología Celular, Universidad Nacional Autónoma de México
Genética Molecular
Ciudad Universitaria
México, D.F. 4510
Mexico

Dear Dr. Pérez-Martínez,

Thank you for submitting your manuscript entitled "The cytochrome b carboxyl-terminal region is necessary for mitochondrial Complex III assembly" to Life Science Alliance. The manuscript was assessed by expert reviewers, whose comments are appended to this letter. We invite you to submit a revised manuscript addressing the Reviewer comments besides the request of reviewer 3 to run the experiment using the Cyb mutant still bearing the complete C-terminal region, as pointed out by Reviewer 2 in their cross-commenting section.

Thank you for this interesting contribution to Life Science Alliance. We are looking forward to receiving your revised manuscript.

Sincerely,

B. MANUSCRIPT ORGANIZATION AND FORMATTING:

Reviewer #1 (Comments to the Authors (Required)):

LSA-2022-01858-T, The cytochrome b carboxyl-terminal region ...

The respiratory complex-III is composed of ten subunits in yeast, but only its heme-b cytochrome is mitochondrially encoded and assembles with its nuclear counterparts in a modular process which eventually leads to the formation of respiratory supercomplexes.

Using two short truncations in the C-terminus of cyt b also in the background of several assembly mutants, the authors conclude that this short terminal region, while dispensable for efficient translation, is critically important for subsequent assembly steps and for regulation. In particular, complexome profiling has successfully been employed to assess the remaining, or disturbed interactions in the various intermediates.

The ms is well presented, and results are conclusively discussed.

Reviewer #2 (Comments to the Authors (Required)):

In the yeast *S. cerevisiae*, the expression of several mitochondrion-encoded genes is regulated at the translational level by the action of translational factors that coordinate protein synthesis with OXPHOS complex assembly. This is the case of Cytb, a catalytic subunit of complex III, which requires the proteins Cbp3 and Cbp6 for its efficient synthesis and hemylation. Here, Flores-Mireles and co-authors investigate the role of Cytb C-terminal region in this process. The manuscript convincingly shows that the last 8 amino acids of Cytb play a critical role in the assembly-feedback regulatory mechanism, likely destabilizing Cbp3-6 interaction with Cytb. The truncated Cytb polypeptide, although synthesized normally, is unstable and does not accumulate in the fully assembled complex III, but in early assembly intermediates which composition was comprehensively investigated by complexome profiling. Notably, C-terminal truncation of Cytb affects its interaction with the early assembly complex III subunit Qcr7. Overall, the work provides novel information regarding the regulation of CytB expression in yeast mitochondria and expand our understanding of the early stages of complex III assembly. Since the hemylation status of CytB is not directly investigated, the discussion about heme b presence/insertion in the different assembly intermediates remains speculative and in my opinion the conclusion that the CytB C-terminus is crucial to regulate CytB hemylation is not supported by experimental evidences.

Minor points:

- The decrease in newly synthesized Cytb immunoprecipitated with Cbp3-HA, as quantified in figure 2B, is statistically significant only for CytbDeltaC8. The text should reflect the statistical analysis outcome.

Referee Cross-Commenting

In reference to reviewer 3 comments:

- In my opinion, in organello labelling and IP provides a more compelling result supporting Cpb3 interaction with newly synthesized CytB than an in whole cells approach. Protein extraction under native conditions requires enzymatic digestion of the yeast cell wall, a step during which cells remain metabolically active and newly synthesized proteins proceed into the assembly pathway.
- Although a CytB mutant bearing the C-terminus but lacking other functional regions could be of interest, in my opinion it is not essential to support the model proposed by the authors. In addition, this is a major and time-consuming task since it requires mitochondrial transformation.

Reviewer #3 (Comments to the Authors (Required)):

Flores-Mireles and colleagues have investigated the role of the C-terminal region of cytochrome b in its assembly in respiratory complex III in yeast, with particular attention to the coordinated regulation of protein synthesis and assembly of the functional complex. This is an important issue in the study of cell metabolism, since all the respiratory complexes but complex II comprise both components encoded by the nuclear genome and the mitochondrial one. In recent years, respiratory complex III assembly is emerging as a key process, since complex III itself acts as a platform for the assembly of complex I and IV (DOI: 10.15252/embj.2019102817). Interestingly, this is the only complex containing only one mitochondrial genome-encoded component.

The present work is therefore of interest to the community, and presents interesting data, that appear rather preliminary, though.

Specific comments:

- Autoradiography shown in Fig.1 should be somehow normalized. Since the authors report having transferred the gels onto membranes before exposure, the same membranes can be probed for immunodetection of mitochondrial proteins (e.g., CS, as they have done for the IP experiments shown in Fig. 2).
- It is unclear why the authors performed in vivo labeling of mitochondrial products in the experiments shown in Fig.1, and then in organello translation (on isolated mitochondria) for the co-immunoprecipitation experiments shown in Fig. 2. An in vivo labeling, consistent with the preliminary data on translation rate of the different Cyb mutant, would have been more reliable. Moreover, the IP experiment only detects interaction between Cbp3 and the newly synthesized proteins, and the membrane should have been probed for Cytb too, in order to verify the interaction at the steady state, in light of the subsequent results showing the presence of assembly intermediates containing both Cytb C13 and Cbp3-6.
- The Cytb-Cbp3-6 binding is disrupted when the complex is fully assembled. Therefore, we should expect higher amount of free Cbp3-6 when Cytb is prevalently detected in CIII2. However, in WT cells, free Cbp3-6 are hardly detectable (Fig. EV2). Can the authors comment on this?
- Similarly, reduced Cbp3-Cytb C13 binding and no Qcr7-Cytb C13 binding should correspond to increased ribosome-bound Cbp3-6 and increased protein synthesis rate of Cytb C13. In other words, according to the model hypothesizing competition between ribosome-binding and Cytb-binding for Cbp3-6, low Cbp3-Cytb C13 binding should correspond to increased Cytb C13 translation. However, this is not what we see in Fig. 1. Cbp3/6-ribosome binding in the Cytb C13 strain should be investigated by mitoribosome isolation and Cbp3-6 immunodetection, or equivalent approaches.
- Correlated to the previous point: the authors state that "The increased free fractions and heterodimers of Cbp3/Cbp6 probably accumulated due to their lower affinity to Cytb Δ C13 and thus were released from assembly intermediates, kept Cytb synthesis going, but without proper regulation. This lack of regulation might be due to the presence of increased concentrations of Cbp3/Cbp6 heterodimers or free Cbp3 and Cbp6 in the mutant." This is rather generic: the regulation should be investigated.
- A Cyb mutant still bearing the complete C-terminal region, but devoid of other important regions (e.g.: the catalytic domain) could be used for comparison; is CIII regularly assembled in that case although constituting a nonfunctional complex?
- In Cytb C13, CIII activity is dramatically reduced, but CII and CIV activity is unaffected. What about in vivo metabolic assays? Do these strains completely abrogate respiration?

Minor point:

The "Methods" section does not always provide sufficient details to reproduce the experiments and should be expanded. This includes, but is not limited to, the following:

- in the "Enzyme activity assays" section is unclear how measurements of complex IV activity have been performed, since "all data were corrected against complex IV activities";
- In the "Synthesis of mitochondrial proteins", the concentration and specific activity of the radiolabeled amino acids are not reported;
- Product codes of antibodies are not reported.

Novella Guidi, PhD
Scientific Editor
Life Science Alliance
manuscript #LSA-2022-01858-T

Dear Dr. Guidi,

Attached please find the revised version of the manuscript #LSA-2022-01858-T entitled "The cytochrome b carboxyl-terminal region is necessary for mitochondrial Complex III assembly", by Daniel Flores-Mireles and co-workers. Here are the answers to the points raised only by reviewers #2 and #3, since reviewer #1 did not requested any change to the manuscript.

Best regards,

Dr. Xochitl Perez-Martinez
Departamento de Genética Molecular
Instituto de Fisiología Celular
UNAM
Circuito Exterior s/n, Ciudad Universitaria
CDMX 04510
Mexico

Reviewer #2 (Comments to the Authors (Required)):

In the yeast *S. cerevisiae*, the expression of several mitochondrion-encoded genes is regulated at the translational level by the action of translational factors that coordinate protein synthesis with OXPHOS complex assembly. This is the case of Cytb, a catalytic subunit of complex III, which requires the proteins Cbp3 and Cbp6 for its efficient synthesis and hemylation. Here, Flores-Mireles and co-authors investigate the role of Cytb C-terminal region in this process. The manuscript convincingly shows that the last 8 amino acids of Cytb play a critical role in the assembly-feedback regulatory mechanism, likely destabilizing Cbp3-6 interaction with Cytb. The truncated Cytb polypeptide, although synthesized normally, is unstable and does not accumulate in the fully assembled complex III, but in early assembly intermediates which composition was comprehensively investigated by complexome profiling. Notably, C-terminal truncation of Cytb affects its interaction with the early assembly complex III subunit Qcr7. Overall, the work provides novel information regarding the regulation of CytB expression in yeast mitochondria and expand our understanding of the early stages of complex III assembly.

Since the hemylation status of CytB is not directly investigated, the discussion about heme b presence/insertion in the different assembly intermediates remains speculative and in my opinion the conclusion that the CytB C-terminus is crucial to regulate CytB hemylation is not supported by experimental evidences.

> We agree that our experiments do not support a conclusion regarding a role of the Cytb C-terminus in the hemylation of the subunit. To avoid confusion, we modified the text related to the hemylation regulation in the discussion section.

Minor points:

- The decrease in newly synthesized Cytb immunoprecipitated with Cbp3-HA, as quantified in figure 2B, is statistically significant only for CytbDeltaC8. The text should reflect the statistical analysis outcome.

> The text describing Figure 2B was modified in the results section to emphasize that only mutant CytbDeltaC8 showed statistical significance on IP reduction.

Referee Cross-Commenting

In reference to reviewer 3 comments:

- In my opinion, in organello labelling and IP provides a more compelling result supporting Cpb3 interaction with newly synthesized CytB than an in whole cells approach. Protein extraction under native conditions requires enzymatic digestion of the yeast cell wall, a step during which cells remain metabolically active and newly synthesized proteins proceed into the assembly pathway.
- Although a CytB mutant bearing the C-terminus but lacking other functional regions could be of interest, in my opinion it is not essential to support the model proposed by the authors. In addition, this is a major and time-consuming task since it requires mitochondrial transformation.

> We agree with the reviewer #2 and our answer was added in the respective issue.

Reviewer #3 (Comments to the Authors (Required)):

Flores-Mireles and colleagues have investigated the role of the C-terminal region of cytochrome b in its assembly in respiratory complex III in yeast, with particular attention to the coordinated regulation of protein synthesis and assembly of the functional complex. This is an important issue in the study of cell metabolism, since all the respiratory complexes but complex II comprise both components encoded by the nuclear genome and the mitochondrial one. In recent years, respiratory complex III

assembly is emerging as a key process, since complex III itself acts as a platform for the assembly of complex I and IV (DOI: 10.15252/emj.2019102817). Interestingly, this is the only complex containing only one mitochondrial genome-encoded component. The present work is therefore of interest to the community, and presents interesting data, that appear rather preliminary, though.

Specific comments:

- Autoradiography shown in Fig.1 should be somehow normalized. Since the authors report having transferred the gels onto membranes before exposure, the same membranes can be probed for immunodetection of mitochondrial proteins (e.g., CS, as they have done for the IP experiments shown in Fig. 2).

> *The experiment of Figure 1B was repeated and citrate synthase was detected by western blot as loading control.*

- It is unclear why the authors performed in vivo labeling of mitochondrial products in the experiments shown in Fig.1, and then in organello translation (on isolated mitochondria) for the co-immunoprecipitation experiments shown in Fig. 2. An in vivo labeling, consistent with the preliminary data on translation rate of the different Cytb mutant, would have been more reliable.

> *In agreement with reviewer's 2 opinion, due to technical complications derived from protein extraction from whole cells, labeling of purified mitochondria and IP provides a more reliable result. This is relevant for the Cbp3-Cytb interaction, since it is not a particularly strong interaction (Gruschke et al 2011), and mitochondria are more concentrated if we perform in organello translation instead of whole cell labeling.*

Moreover, the IP experiment only detects interaction between Cbp3 and the newly synthesized proteins, and the membrane should have been probed for Cytb too, in order to verify the interaction at the steady state, in light of the subsequent results showing the presence of assembly intermediates containing both Cytb Δ C13 and Cbp3-6.

> *We agree with reviewer 3 that the information about the Cbp3-Cytb interaction is different from in vivo labeling (which reveals interaction with newly synthesized Cytb) and from western blot analysis (which reveals steady state interactions). Unfortunately, our Cytb antibody is not very efficient or sensitive, so even WT Cytb was difficult to detect after Cbp3-HA IP. In Figure for reviewer 1 we show one of our repeats. In this case a faint IP band of WT Cytb is observed while nothing is detected in Cytb Δ C8 and Cytb Δ C13 mutants after IP. Even though this result is consistent with the figure*

presented on the manuscript, the signals are so weak that we did not think we can withdraw any conclusion from this experiment, so we preferred to leave it out.

> Figure legend for Fig for Reviewer 1: Mitochondria (500 µg) from the indicated cells were solubilized with 1% digitonin and the cleared extract was incubated with an anti-HA antibody to immunoprecipitate Cbp3. Total (representing 5% of the samples before solubilization) and immunoprecipitated (IP) fractions were analyzed by western blot with the indicated antibodies. Cs is an antibody against citrate synthase, used as a control for no expected Cbp3 interaction.

• The Cytb-Cbp3-6 binding is disrupted when the complex is fully assembled. Therefore, we should expect higher amount of free Cbp3-6 when Cytb is prevalently detected in CIII2. However, in WT cells, free Cbp3-6 are hardly detectable (Fig. EV2). Can the authors comment on this?

> Cbp3/Cbp6 association with Cytb is a dynamic process. We believe that in WT cells Cbp3-Cbp6 is mainly engaged on Cytb assembly, and therefore hardly detected as a free fraction. In contrast, when Cytb accumulation is reduced due to a mutation, the Cbp3/Cbp6 is enriched as a free heterodimer. This is not the first report of such observation. In Gruschke et al, 2012, Figure 5A the same is observed. The only way to clearly detect the free Cbp3/Cbp6 heterodimer was in mutant conditions, like when this group used a strain with ectopically expressed Cytb (Figure 4A), in which case there is a reduced accumulation of Cytb.

• Similarly, reduced Cbp3-Cytb Δ C13 binding and no Qcr7-Cytb Δ C13 binding should correspond to increased ribosome-bound Cbp3-6 and increased protein synthesis rate of Cytb Δ C13. In other words, according to the model hypothesizing competition between ribosome-binding and Cytb-binding for Cbp3-6, low Cbp3-Cytb Δ C13 binding should correspond to increased Cytb Δ C13 translation. However, this is not what we see in Fig. 1. Cbp3/6-ribosome binding in the Cytb Δ C13 strain should be investigated by mitoribosome isolation and Cbp3-6 immunodetection, or equivalent approaches.

> The current model establishes that there is a competition between Cbp3/Cbp6 in its free form and Cytb-bound form, this agrees with our observation that the Cbp3/Cbp6 heterodimer is enriched in its free form in the CytbDeltaC13 mutant. Indeed, Gruschke et al (2012) observed that overexpression of Cbp3/Cbp6 increased free fractions and Cytb synthesis. Salvatori et al (2020) demonstrated that synthesis of Cytb depends on free Cbp3/Cbp6 fractions. Interestingly, the role of ribosome-bound Cbp3/Cbp6 to the large subunit is not completely understood. Following the reviewer's suggestion, we

performed sucrose cushion ultracentrifugation of mitochondrial extracts from WT and CytbDeltaC13 cells to isolate mitoribosomes. We observed no clear difference in the amount of Cbp3 that pelleted with the isolated mitoribosomes. This result is shown in new Figure 1E.

- Correlated to the previous point: the authors state that "The increased free fractions and heterodimers of Cbp3/Cbp6 probably accumulated due to their lower affinity to Cytb Δ C13 and thus were released from assembly intermediates, kept Cytb synthesis going, but without proper regulation. This lack of regulation might be due to the presence of increased concentrations of Cbp3/Cbp6 heterodimers or free Cbp3 and Cbp6 in the mutant." This is rather generic: the regulation should be investigated.

> We agree that many questions still remain elusive regarding the Cytb synthesis regulation. Indeed, in Garcia-Guerrero et al (2018) we proposed the presence of two unknown additional factors involved in such regulation, which could be related (or not) to the presence of the Cytb C-terminus. However, we respectfully think that these studies are beyond the aim of the present study and would require long time-consuming experiments and validation.

- A Cytb mutant still bearing the complete C-terminal region, but devoid of other important regions (e.g.: the catalytic domain) could be used for comparison; is CIII regularly assembled in that case although constituting a nonfunctional complex?

> This interesting inquiry has been previously tackled. Hindenbeutel et al (2014) mutated both Cytb heme binding sites and observed that lack of hemylation reduced Cytb synthesis due to sequestration of Cbp3/Cbp6 in early assembly subcomplexes. In addition, in agreement with Reviewer's 2 opinion to mutate additional sites (other than the catalytic heme binding sites) would not provide support to our current model regarding the role of the Cytb C-terminus on Cytb and complex III biogenesis.

- In Cytb Δ C13, CIII activity is dramatically reduced, but CII and CIV activity is unaffected. What about in vivo metabolic assays? Do these strains completely abrogate respiration?

*> Upon truncation of the Cytb C-terminus, biogenesis of CIII cannot proceed and no CIII is thus fully assembled. The absence of CIII was demonstrated by different biochemical approaches and complexome profiling. The residual CIII activity observed in Fig EV3B is most likely noise. Since *S. cerevisiae* yeast cells do require a functional CIII to*

transfer electrons to cytochrome c and CIV is the only one terminal oxidase, it was not surprising that our mutant is absolutely unable to grow on respiratory media. The strain is, however, fully capable to grow with fermentable carbon sources, such as glucose and galactose. We expanded panel Fig 1A to show this. We consider that performing in vivo metabolic assays will not reflect major findings since our mutant has zero respiration and the subsequent metabolic and complexome remodeling are beyond the scope of our paper.

Minor point:

The "Methods" section does not always provide sufficient details to reproduce the experiments and should be expanded. This includes, but is not limited to, the following:

- in the "Enzyme activity assays" section is unclear how measurements of complex IV activity have been performed, since "all data were corrected against complex IV activities";

> Thanks for pointing this out, the activities were actually normalized against CII (and not CIV) as it is described in the legend of Supplementary Figure EV3 (now Supplementary Figure 3). Our apologies, we have amended the respective methods section.

- In the "Synthesis of mitochondrial proteins", the concentration and specific activity of the radiolabeled amino acids are not reported;

> This information is now incorporated in the Materials section.

- Product codes of antibodies are not reported.

> The code for the commercial antibodies used in the present study were added to the Materials section.

April 3, 2023

RE: Life Science Alliance Manuscript #LSA-2022-01858-TR

Dr. Xochitl Pérez-Martínez
Universidad Nacional Autónoma de México
Instituto de Fisiología Celular. Genética Molecular
Ciudad Universitaria
México, D.F. 4510
Mexico

Dear Dr. Pérez-Martínez,

Thank you for submitting your revised manuscript entitled "The cytochrome b carboxyl-terminal region is necessary for mitochondrial Complex III assembly". We would be happy to publish your paper in Life Science Alliance pending final revisions necessary to meet our formatting guidelines.

- please address the final Reviewer 3's point
- please add ORCID ID for both corresponding authors-you should have received instructions on how to do so
- please add the Twitter handle of your host institute/organization as well as your own or/and one of the authors in our system
- please make sure that the author names in the manuscript match the author names entered in our system

Figure Check:

*Figure 1D (left part) is duplicated in Figure S1 B (left part): this is not allowed, so please remove the duplication and substitute with another blot

A. FINAL FILES:

B. MANUSCRIPT ORGANIZATION AND FORMATTING:

Sincerely,

Reviewer #3 (Comments to the Authors (Required)):

In response to the Reviewers' criticisms, the Authors have performed some new experiments, amended some inaccuracies in the manuscript, and discussed limitations. Overall, in my opinion, the manuscript is improved after revisions. Although I acknowledge that, as underlined by Reviewer 2 and the Authors themselves, some of my previous comments required time-consuming experiments to be more realistically tackled in subsequent works, some of my concerns still stand. However, the Authors have investigated a complex issue, that is expected to be solved by evidence accumulating in little steps. Therefore, I believe that this manuscript deserves publication, once the limitations of the study have been clearly acknowledged. In this regard, I would recommend including statements, in the appropriate points of the discussion, about the "still elusive" mechanisms that remain to be elucidated, especially as far as the Cytb synthesis regulation is concerned.

Thank you for submitting your revised manuscript entitled "The cytochrome b carboxyl-terminal region is necessary for mitochondrial Complex III assembly". We would be happy to publish your paper in Life Science Alliance pending final revisions necessary to meet our formatting guidelines.

-please address the final Reviewer 3's point

> *Answers to Reviewer 3 are below.*

-please add ORCID ID for both corresponding authors-you should have received instructions on how to do so

> *We have added ORCID ID from corresponding authors*

-please add the Twitter handle of your host institute/organization as well as your own or/and one of the authors in our system

> *We have added the Twitter handle of IFC UNAM*

-please make sure that the author names in the manuscript match the author names entered in our system

> *We have checked this point.*

Figure Check:

*Figure 1D (left part) is duplicated in Figure S1 B (left part): this is not allowed, so please remove the duplication and substitute with another blot

> *We apologize, Figure S1B is part of the same experiment from Figure 1D, so that is why we repeated the WT lane. However, we replaced figure S1B with a different repeat.*

Reviewer #3 (Comments to the Authors (Required)):

In response to the Reviewers' criticisms, the Authors have performed some new experiments, amended some inaccuracies in the manuscript, and discussed limitations. Overall, in my opinion, the manuscript is improved after revisions.

Although I acknowledge that, as underlined by Reviewer 2 and the Authors themselves, some of my previous comments required time-consuming experiments to be more realistically tackled in subsequent works, some of my concerns still stand. However, the Authors have investigated a complex issue, that is expected to be solved by evidence accumulating in little steps. Therefore, I

believe that this manuscript deserves publication, once the limitations of the study have been clearly acknowledged. In this regard, I would recommend including statements, in the appropriate points of the discussion, about the "still elusive" mechanisms that remain to be elucidated, especially as far as the Cytb synthesis regulation is concerned.

> In agreement with reviewer 3 suggestion, we added to the Discussion section (page 13) a phrase regarding the role of different Cytb regions on COB mRNA translation:

It remains elusive how different regions of the Cytb protein (like soluble loops or transmembrane specific residues) could exert regulatory roles on COB mRNA translation. Interestingly, mutation of the catalytic heme b binding sites on Cytb decreased synthesis (Hildenbeutel et al., 2014), while deletion of the last 8-13 residues from the Cytb C-terminus released translational regulation.

> Regarding the role of Cbp3/Cbp6 population that is bound to the mitoribosome, the next phrase (Results section, page 6) was added:

Cbp3 and Cbp6 are associated with the mitoribosome (Gruschke et al, 2011), although the exact role of this Cbp3/Cbp6 population on translation is not well understood.

April 11, 2023

RE: Life Science Alliance Manuscript #LSA-2022-01858-TRR

Dr. Xochitl Pérez-Martínez
Universidad Nacional Autónoma de México
Instituto de Fisiología Celular. Genética Molecular
Ciudad Universitaria
México, D.F. 4510
Mexico

Dear Dr. Pérez-Martínez,

Thank you for submitting your Research Article entitled "The cytochrome b carboxyl-terminal region is necessary for mitochondrial Complex III assembly". It is a pleasure to let you know that your manuscript is now accepted for publication in Life Science Alliance. Congratulations on this interesting work.

DISTRIBUTION OF MATERIALS:

Again, congratulations on a very nice paper. I hope you found the review process to be constructive and are pleased with how the manuscript was handled editorially. We look forward to future exciting submissions from your lab.

Sincerely,
